# Cooperative Standoff Target Tracking Using Multiple Fixed-Wing UAVs with Input Constraints in Unknown Wind

Zhong Liu [1,*], Lingshuang Xiang [2] and Zemin Zhu [1]

1 School of Computer Science, Huanggang Normal University, Huanggang 438000, China; jsjzzm@hgnu.edu.cn
2 ChangJiang Industry Investment Group Co., Ltd, Wuhan 430062, China; xianglingshuang@cjchanye.com
* Correspondence: liuzhong@hgnu.edu.cn; Tel.: +86-158-2973-2829

**Abstract:** This paper investigates the problem of cooperative standoff tracking using multiple fixed-wing unmanned aerial vehicles (UAVs) with control input constraints. In order to achieve accurate moving target tracking in the presence of unknown background wind, a coordinated standoff target tracking algorithm is proposed. The objective of the research is to steer multiple UAVs to fly a circular orbit around a moving target with prescribed intervehicle angular spacing. To achieve this goal, two control laws are proposed, including relative range regulation and space phase separation. On one hand, a heading rate control law based on a Lyapunov guidance vector field is proposed. The convergence analysis shows that the UAVs can asymptotically converge to a desired circular orbit around the target, regardless of their initial position and heading. Through a rigorous theoretical proof, it is concluded that the command signal of the proposed heading rate controller will not violate the boundary constraint on the heading rate. On the other hand, a temporal phase is introduced to represent the phase separation and avoid discontinuity of the wrapped space phase angle. On this basis, a speed controller is developed to achieve equal phase separation. The proposed airspeed controller meets the requirements of the airspeed constraint. Furthermore, to improve the robustness of the aircraft during target tracking, an estimator is developed to estimate the composition velocity of the unknown wind and target motion. The proposed estimator uses the offset vector between the UAV's actual flight path and the desired orbit, which is defined by the Lyapunov guidance vector field, to estimate the composition velocity. The stability of the estimator is proved. Simulations are conducted under different scenarios to demonstrate the effectiveness of the proposed cooperative standoff target tracking algorithm. The simulation results indicate that the temporal-phase-based speed controller can achieve a fast convergence speed and small phase separation error. Additionally, the composition velocity estimator exhibits a fast response speed and high estimation accuracy.

**Keywords:** multiple UAVs; cooperative control; target tracking; Lyapunov guidance vector field; unknown background wind

## 1. Introduction

Tracking a moving ground target is one of the important capabilities of UAVs [1]. Making the tracking process automatic and free of human intervention is essential for relieving the burden on UAV operators and improving the efficiency and safety of UAV missions. The goal of this paper is to develop a control scheme that allows multiple fixed-wing UAVs to cooperatively track a moving ground target in unknown windy conditions.

Target tracking using multiple fixed-wing UAVs remains a challenge. On one hand, the motion of fixed-wing UAVs is subject to various input constraints. To prevent stalling, a fixed-wing aircraft cannot hover and must maintain a positive forward airspeed. Accordingly, the UAVs must fly in a circle around the ground-based moving target [2]. On the other hand, to avoid collisions and to ensure that the sensors can cover the target, the UAVs need to be evenly distributed around the target and maintain a certain phase interval [3]. Standoff tracking is a possible solution for target tracking using a team of fixed-wing UAVs.

In this pattern, the UAVs keep a certain distance (called the standoff radius) from the target and move in a circular motion (termed the standoff circle) at a proper altitude relative to the target.

In cooperative target tracking missions with unknown wind, three main technical issues should be considered: (1) Relative distance regulation, which focuses on how to enable the UAVs to converge to a circular orbit with a prescribed standoff radius around the target by controlling their headings [4]. (2) Intervehicle phase separation, which focuses on distributing the hovering UAVs uniformly over a standoff circle with a certain angular phase difference by controlling their airspeeds [5]. (3) Background wind resistance, which focuses on how to achieve robust stability for UAVs performing standoff tracking in the presence of wind and the target's motion [6].

Relative distance regulation, which aims to steer the UAV to a circular orbit around the target, is the key to achieving standoff tracking of a moving target with a single UAV. Typically, a guidance law is proposed to regulate the position of the UAV on a predefined circular path. The path usually guides the UAV to circle around a ground-based moving target at a constant distance. Consequently, the UAV trajectory can be expressed as a circle with a predefined standoff radius in the target's frame. There are various types of the guidance law, including reference point guidance (RPG) [7–9], Lyapunov guidance vector field (LGVF) [10–14], and so on. Based on a predefined target tracking path, the standoff tracking problem of a ground-based moving target can be converted into a path following problem [7]. For example, in [8,9], a nonlinear guidance law is proposed to achieve path following for a curved path. In this approach, each point on the curved path is designated as a reference point, and a lateral acceleration command is generated to drive the UAV to the reference point. However, because the ground-based target's moving speed is usually much slower than that of the UAV, the RPG method cannot be directly applied to the standoff target tracking problem. As a new form of potential field, the LGVF is introduced to guide the UAVs to achieve standoff target tracking. In [10], an LGVF is proposed for hovering maneuvers around a stationary target. This approach also enables moving target tracking, but it may lead to slow convergence due to constant curvature in the LGVF. To shorten the convergence time, the authors in [11] combine the tangent with the LGVF, while in [12,13], the authors add the circulation parameter $c$ into the original LGVF. The shape of the LGVF can be adaptively adjusted by changing the circulation parameter. In this way, a faster convergence to the standoff circle can be achieved due to a higher contraction component. Based on the works of [12,13], an offline optimal parameter searching method was presented for selecting the optimal guidance function to shorten the convergence time in [14]. However, this approach has such a heavy computational load that it cannot be extended to real-time application scenarios. Although these above methods have been verified to be feasible for the single UAV standoff tracking problem, they only focus on the optimization of the LGVF without considering the input constraints of fixed-wing UAVs, such as heading rate limitations. If the curvature of the LGVF is too large, the actual trajectory of a UAV cannot converge to a standoff circle due to the saturated rudders. Therefore, it is still necessary to design a control law for regulating the relative distance while satisfying the turning rate limitation.

The performance of target localization algorithms is significantly impacted by the relative sensor-target geometry. Observation configurations, or different sensor-target geometries, produce varying uncertainty ellipses of the target location algorithm. It is worth considering which observation configuration can yield the best target localization results. By minimizing the Cramer–Rao lower bound (CRLB), which provides a lower bound on the estimator performance, the uncertainty in the estimation process can be reduced. Therefore, in [15] they utilize the determinant of the CRLB to determine the observation configuration that results in a minimal measure of the uncertainty ellipse. According to the conclusions in [15], if only two UAVs perform a standoff tracing mission, the intersection angle subtended at the target (called the phase separation angle) by two UAVs will be $\left(\frac{\pi}{2}\right)$; if the number of UAVs is $N \geq 3$, the phase separation angle between

adjoining UAVs will be $\left(\frac{2\pi}{N}\right)$. An optimal configuration should position the UAVs at equal angular intervals around the perimeter of the standoff circle. This requires phase separation in the coordinated standoff tracking problem. Various phase separation methods have been proposed, including model predictive control (MPC) [16,17], sliding mode control (SMC) [18,19], conical pendulum motion [20], consensus algorithm [21–24], and so on. In [16,17], a nonlinear MPC framework for coordinated standoff tracking by two UAVs is proposed The optimal control outputs for speed and turning rate are generated by minimizing the sum of weighted cost functions that include the standoff-distance regulation error and phase separation error in a receding horizon. However, as the number of UAVs increases, the computational complexity and iterative time required for searching optimal results also increase. In [18], a hovering algorithm based on sliding mode control is presented to control the virtual leader's position on a standoff circle centered at the ground-based moving target. However, the SMC method suffers from chattering due to the discontinuity of the signum function in the control law. To eliminate chattering, the signum function is replaced by a saturation function in Ref. [19]. However, both Refs. [18,19] ignore the vehicle airspeed constraint. The airspeed of the fixed-wing UAV is restricted within lower and upper bounds. In order to satisfy the airspeed boundary constraint, Ref. [20] proposes that the UAV reduces its speed by decreasing the standoff radius when flying on the right-hand side of the target and its airspeed reaches the upper bound. Conversely, the UAV increases the standoff radius to increase its airspeed when its airspeed reaches the lower bound. However, this approach causes the distance between the UAV and the target to oscillate, leading to failure in achieving the control objectives of the cooperative standoff target tracking problem. Different from the integrated controller in [20], the heading control channel and velocity control channel are decoupled in [21–24]. The space phase angle is chosen as the coordination variable of the consensus algorithm, allowing for the design of cooperative airspeed controllers for the UAVs. The phase separation angles among the UAVs can asymptotically converge, reflecting equal space separation. However, the airspeed constraint is not taken into account in the space phase separation method. Furthermore, the phase separation angles between $[-\pi, \pi)$ are discontinuous, and will lead to oscillating airspeed control input. This is not beneficial for reducing the space phase separation error. Therefore, it is critical to design a phase separation controller that provides a smooth signal output and meets the airspeed constraint of UAVs in the coordinated standoff tracking problem.

Most of the research mentioned above assumes an ideal non-wind environment or a known constant background wind. However, in practice, background wind is usually present and it affects the performance of the UAVs, especially when the background wind is unknown. In [25], a robust term is added to the standoff tracking control law to obtain disturbance rejection for wind gusts. However, the response time of the robust controller is too long and unsuitable for target tracking requiring high maneuverability. To quantitatively describe wind dynamics, a simple conservative model (e.g., sine function as in [26] or linear model as in [27]) or a more sophisticated one (e.g., stochastic as in [28]) can be used. In [28], the Dryden model [29] and Davenport model [30] are used to describe the dynamics of wind at high altitudes or near the ground, respectively. The unscented Kalman filter (UKF) is introduced to estimate the velocity of background wind. However, the accuracy of the wind speed estimation largely depends on the accuracy of the constructed wind dynamic model. In reality, wind is stochastic and time-varying, which makes it difficult to model. In [31], an adaptive estimator is utilized to estimate the wind velocity. In the case of a stationary target without wind, the vehicle trajectory converges to a circular orbit (called a perfect trajectory or orbit) by implementing a designed standoff tracking control law. However, for a moving target with wind, the actual trajectory cannot converge to the perfect one due to the disturbance caused by the wind and the target's motion. The wind velocity estimation can be obtained by reducing the offset between the actual vehicle trajectory and the perfect trajectory. However, the convergence rate of the estimator is slow

because it only uses the radial distance of the offset. Therefore, designing a wind velocity estimator that has a high accuracy and fast convergence rate remains challenging.

This paper addressed the challenges of cooperative standoff target tracking using multiple fixed-wing UAVs with input constraints and the considerations of unknown background wind and target motion. Controllers satisfying the input constraints, such as heading rate and airspeed limitation, are designed to guarantee a team of fixed-wing UAVs can perform efficiently during coordinated standoff tracking missions. The major contributions of the paper are as follows: (a) A heading rate control law based on LGVF is proposed. To satisfy the limitation of the heading rate, the minimum allowable standoff radius is formulated. It is proved that this proposed heading rate controller can guarantee that the UAV can asymptotically converge to a standoff circle hovering over the target under arbitrary initial conditions of the position and heading. (b) To avoid discontinuities in the space phase angle, a new term called the temporal phase is proposed to represent the phase separation. An airspeed control law is introduced to steer a team of UAVs maintaining an optimal observation configuration, which requires distribution around the standoff circle with equal phase separation. The proofs for satisfying the airspeed limitation and global convergence using the proposed speed controller are provided. (c) The target's motion and background wind are regarded as external disturbances. The offset vector caused by external disturbances between the actual trajectory and the perfect/desired orbit is utilized to estimate the composition velocity of wind and the target's motion. It is proved that the estimated result asymptotically converges to the true value of the composition velocity.

The remainder of this paper is structured as follows. Section 2 presents the problem formulation, including assumptions made in this study, a UAV kinematic model with control input constraints, and control objectives. Section 3 discusses the problem of standoff tracking using a single UAV. Based on the LGVF, a lateral controller with a heading rate input constraint is proposed to regulate the position of a UAV on a circular orbit around the target under the condition of an arbitrary initial position and heading. In Section 4, we introduce a term called temporal phase to represent the spatial distribution of the UAVs on the standoff circle and propose cooperative controllers with an airspeed input constraint to achieve the desired temporal phase separation. In Section 5, an online estimator is developed to adapt the proposed heading rate and airspeed controllers to the case of a moving target in the presence of wind. Section 6 presents a more detailed control and coordination architecture of standoff target tracking. The computational complexity and in-vehicle communication are analyzed in more detail. Simulation and experimental results are demonstrated in Section 7, followed by a summary and conclusions in Section 8.

## 2. Problem Formulations

Without loss of generality, the following assumptions are made to render the problem simpler and well posed.

**Assumption 1:** *The UAVs fly at a constant altitude, and the target moves in a two-dimensional plane, ignoring its height.*

**Assumption 2:** *The position of the moving target is assumed to be known.*

**Assumption 3:** *The communication between the UAVs is ideal, without any restrictions such as limited communication range, packet loss, and delay.*

**Assumption 4:** *The UAV is equipped with a low-level autopilot that holds constant altitude, and follows the command inputs of the speed and heading rate.*

**Remark 1:** *It is assumed that the true value of the target's position is known, and this is used to estimate the composition velocity of the target's motion and background wind. However, the velocities of the target and wind are unknown in this paper. We keep the target's location out of the scope since we aim at providing a solid formulation concerning the problem of coordinated standoff tracking of a ground target. The next endeavor of this work should explore the possibility of using an onboard observation sensor such as a camera to facilitate tracking.*

### 2.1. UAV Model

Under the above assumptions, the inertial frame in a two-dimensional plane is constructed. The *x*- and *y*-axes point east and north, respectively. As shown in Figure 1, the kinematic of the UAV is expressed as follows:

$$\dot{x} = v_s cos\psi + w_x; \dot{y} = v_s sin\psi + w_y; \dot{\psi} = u \tag{1}$$

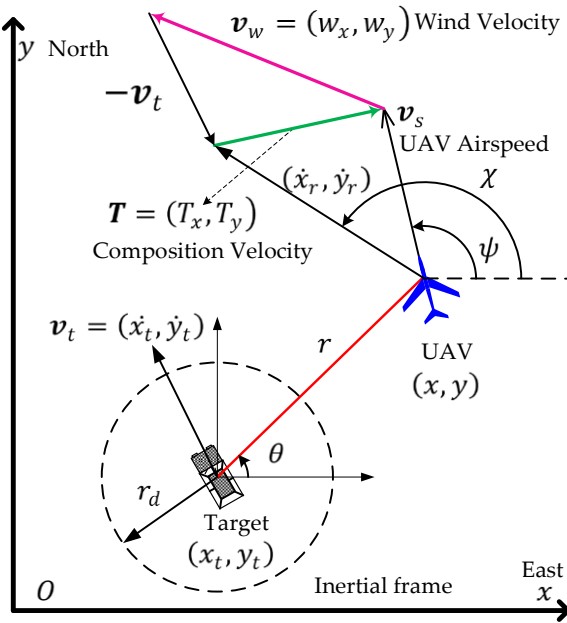

**Figure 1.** Geometry of ground target tracking in wind.

In Equation (1), $v_w = (w_x, w_y)$ is the background wind velocity. $(x, y) \in R^2$ is the two-dimensional position of the UAV in the inertial frame. $\psi \in [-\pi, \pi)$ is the UAV heading. $v_s$ is the true air speed (TAS) of the UAV. *u* is the heading rate of the UAV. $U = (v_s, u)^T$ is the control input signal followed by the low-level autopilot of the UAV. The airspeed and the heading rate should be enforced with the following input constrains.

$$0 < v_{min} \leq v_s \leq v_{max} \tag{2}$$

$$|u| \leq \omega_{max} \tag{3}$$

In Equations (2) and (3), $v_{min}$ and $v_{max}$ are the minimum and maximum airspeed, respectively. $\omega_{max}$ is the upper bound of the heading rate.

### 2.2. The Ground-Based Moving Target Model

The ground-based moving target (GMT) is regarded as a mass point whose position, velocity, and acceleration with respect to the inertial frame are denoted by $(x_t, y_t)$, $(\dot{x}_t, \dot{y}_t)$, and $(\ddot{x}_t, \ddot{y}_t)$, respectively. Concerning the target, we define only its state variables without providing any further information regarding its kinematic model. Thus, our approach covers the general case of a ground-based target's motion.

### 2.3. The Relative Motion Model

The relative motion with background wind is expressed as follows:

$$\dot{x}_r = v_s cos\psi + w_x - \dot{x}_t; \dot{y}_r = v_s sin\psi + w_y - \dot{y}_t; \dot{\psi} = u \tag{4}$$

The target's motion and wind are regarded as external disturbances. These two velocities can be combined into a single velocity term, which is called the composition

velocity and denoted as $T = (T_x, T_y) = (\dot{x}_t - w_x, \dot{y}_t - w_y)$. Equation (4) can be rewritten as follows:

$$\dot{x}_r = v_r cos\chi; \dot{y}_r = v_r sin\chi; \dot{\chi} = \lambda_u(\psi)u \tag{5}$$

In Equation (5), $\chi \in [-\pi, \pi)$ is the relative course angle. $v_r$ is the relative speed. $\lambda_u(\psi)$ is the parameter of relative motion in the model. They are calculated as follows:

$$v_r = \sqrt{v_s^2 + T_x^2 + T_y^2 - 2v_s(T_x cos\psi + T_y sin\psi)} \tag{6}$$

$$\chi = arctan\left(\frac{v_s sin\psi - T_y}{v_s cos\psi - T_x}\right) \tag{7}$$

$$\lambda_u(\psi) = \frac{v_s^2 - v_s(T_x cos\psi + T_y sin\psi)}{v_r^2} \geq \frac{1}{2} \tag{8}$$

The relative motion, as shown in Equation (5), can be expressed in polar coordinates as

$$\begin{bmatrix} \dot{r} \\ r\dot{\theta} \end{bmatrix}_{UAV} = \begin{bmatrix} \dot{x}_r cos\theta + \dot{y}_r sin\theta \\ -\dot{x}_r sin\theta + \dot{y}_r cos\theta \end{bmatrix} = \begin{bmatrix} v_r cos(\chi - \theta) \\ v_r sin(\chi - \theta) \end{bmatrix} \tag{9}$$

The distance vector between the UAV and the target is $r = (x_r, y_r)$, the relative distance is $r = \|r\|$, and the observation phase is $\theta \in [-\pi, \pi)$.

### 2.4. The Objectives of the Control Problem

As shown in Figure 2, there is a team of UAVs ($A_i$, $i = 1, 2, \ldots, N$) performing a cooperative standoff tracking mission for a ground-based moving target in unknown background wind. In the process of tracking, on one hand, each UAV needs to circle around the target to maintain a constant relative distance. On the other hand, the hovering UAVs are distributed around the target with equal phase separation to avoid collisions and maximize the coverage of sensors.

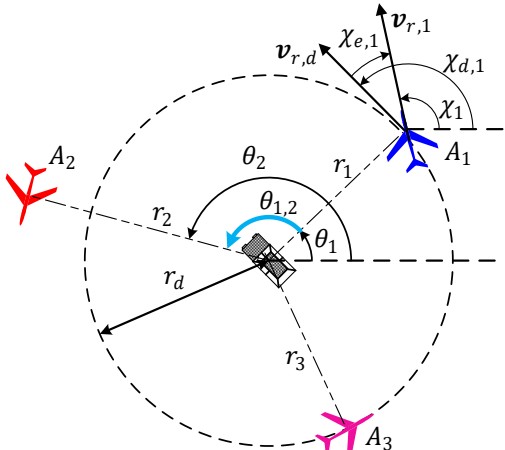

**Figure 2.** Illustration of cooperative standoff target tracking.

The following three control objectives need to be achieved for the multi-UAV cooperative standoff tracking control problem.

(i)   Control objective 1: relative distance regulation. The relative distance $r_i$ between the UAV $A_i$ and the target should be converged to the desired standoff radius $r_d$.

$$\Delta r_i = r_i - r_d \longrightarrow 0 \tag{10}$$

(ii)　Control objective 2: relative course convergence. The relative course $\chi_i$ of the UAV $A_i$ should be converged to the desired relative course $\chi_{d,i}$ to maintain circular motion around the target.

$$\chi_{e,i} = \chi_i - \chi_{d,i} \longrightarrow 0 \tag{11}$$

(iii)　Control objective 3: intervehicle phase separation. The phase separation $\theta_{i,j}$ between $A_i$ and its neighbor $A_j$ should be converged to the optimal configuration $\theta_d$.

$$\Delta\theta_i = \theta_{i,j} - \theta_d = \theta_j - \theta_i - \theta_d \longrightarrow 0 \tag{12}$$

According to Ref. [15], the optimal observation configuration $\theta_d$ is defined as follows:

$$\theta_d = \begin{cases} \pi/2, N = 2 \\ 2\pi/N, \ N > 2 \end{cases} \tag{13}$$

Therefore, when studying the multi-UAV cooperative standoff tracking control problem, the following three issues are mainly considered in this paper.

(a)　How to design the lateral control law subject to the heading rate constraint (Equation (3)) to regulate the relative course of $A_i$ so that $A_i$ can fly along the standoff circle with radius $r_d$ around the target. This would mean that objectives 1 and 2 would both be achieved.

(b)　How to design the longitudinal control law subject to the airspeed limitation (Equation (2)) to guarantee that the hovering UAVs are distributed over the standoff circle with equal phase separation. This would mean that control objective 3 would be achieved.

(c)　How to estimate the composition velocity of wind and target in order to improve the performance of the UAVs during the tracking in the presence of wind and the target's motion.

## 3. Standoff Tracking Using a Single UAV

### 3.1. Guidance Law Based on LGVF

In [10], the desired relative course is generated from an LGVF that guides the UAV to circle around the target with a predefined standoff radius $r_d$. The LGVF for a standoff target tracking can be described as follows:

$$\begin{bmatrix} \dot{x}_r \\ \dot{y}_r \end{bmatrix}_{GF} = -v_r \begin{bmatrix} \left(\frac{x_r}{r}\right)\left(\frac{r^2-r_d^2}{r^2+r_d^2}\right) + \left(\frac{y_r}{r}\right)\left(\frac{2rr_d}{r^2+r_d^2}\right) \\ -\left(\frac{x_r}{r}\right)\left(\frac{2rr_d}{r^2+r_d^2}\right) + \left(\frac{y_r}{r}\right)\left(\frac{r^2-r_d^2}{r^2+r_d^2}\right) \end{bmatrix} \tag{14}$$

Define the following vector field angle $\phi_{GF} \in [0, \pi)$:

$$cos\phi_{GF} = \frac{r_d^2 - r^2}{r^2 + r_d^2}; sin\phi_{GF} = \frac{2rr_d}{r^2 + r_d^2} \tag{15}$$

Equation (15) can be expressed in polar coordinates as follows:

$$\begin{bmatrix} \dot{r} \\ r\dot{\theta} \end{bmatrix}_{GF} = \begin{bmatrix} cos\theta & sin\theta \\ -sin\theta & cos\theta \end{bmatrix} \begin{bmatrix} \dot{x}_r \\ \dot{y}_r \end{bmatrix}_{GF} = v_r \begin{bmatrix} cos\phi_{GF} \\ sin\phi_{GF} \end{bmatrix} \tag{16}$$

The following guidance law is introduced to guide the UAV to fly along the LGVF in Equation (16):

$$\chi_d = \theta + \phi_{GF} \tag{17}$$

where the observation phase $\theta$ is defined by

$$\theta = \begin{cases} arctan\left(\frac{y_r}{x_r}\right), & if \ r > 0 \\ \chi, & if \ r = 0 \end{cases} \tag{18}$$

The feasibility of the guidance law Equation (17) is given by Theorem 1.

**Theorem 1:** *If the guidance law Equation (17) is applied to the relative motion model of the UAV as shown in Equation (9), the relative distance between the UAV and the target asymptotically converges to the predefined standoff radius, i.e., $r \longrightarrow r_d$ as $t \longrightarrow +\infty$.*

**Remark 2:** *The proof of Theorem 1 is seen in [10]. It can be observed from Theorem 1 that in order to achieve standoff tracking, it is required that the relative course $\chi$ should be always aligned with the desired relative course $\chi_d$ generated by the guidance law Equation (17).*

*3.2. Heading Rate Controller Design Subject to the Input Constraints*

According to the conclusion of Theorem 1, if $\chi$ is always equal to $\chi_d$, the UAV eventually converges to the standoff circle with the desired radius $r_d$, and performs the standoff target tracking mission successfully. However, in general, there exists an angle error between $\chi$ and $\chi_d$ initially and the relative course $\chi$ cannot be directly controlled. Thus, we need to design a heading rate controller to guarantee that the relative course $\chi$ eventually converges to $\chi_d$. In other words, the proposed heading rate controller indirectly controls $\chi$ by regulating the UAV heading directly.

The relative course error $\chi_e \in [-\pi, \pi)$ can be defined by

$$\chi_e = \chi - \chi_d \tag{19}$$

Differentiating Equation (17) with respect to $t$ leads to the following equation:

$$\dot{\chi}_d = \dot{\theta} + \dot{\phi}_{GF} \tag{20}$$

According to Equation (9), the dynamic of the azimuth $\theta$ is obtained as

$$\dot{\theta} = \frac{v_r}{r} sin(\chi - \theta) \tag{21}$$

Differentiating Equation (15) with respect to $t$:

$$\dot{\phi}_{GF} = \frac{2r_d}{r^2 + r_d^2} v_r cos(\chi - \theta) \tag{22}$$

The desired relative course rate can be obtained by substituting Equations (21) and (22) into Equation (20).

$$\dot{\chi}_d = \frac{v_r}{r} [sin(\chi_e + \phi_{GF}) + sin\phi_{GF} cos(\chi_e + \phi_{GF})] \tag{23}$$

It can be observed from Equation (23) that when $r \to 0$, $\dot{\chi}_d \to +\infty$. Thus, in order to facilitate the engineering application, let $\dot{\chi}_d = \frac{4v_r}{r_d}$ when $r = 0$. Based on the relative course error $\chi_e$, Equation (19), and the desired relative course rate $\dot{\chi}_d$ Equation (23), the heading rate controller is designed as follows:

$$u = \begin{cases} \omega, & if |\omega| \leq \omega_{max} \\ sgn(\omega)\omega_{max}, & if |\omega| > \omega_{max} \end{cases} ; \omega = -k\chi_e + \frac{\dot{\chi}_d}{\lambda_u(\psi)}; k > 0 \tag{24}$$

It is observed from Equation (24) that the heading rate controller consists of two main parts: a feedback term $\left(\frac{\dot{\chi}_d}{\lambda_u(\psi)}\right)$ and a feedforward term $(-k\chi_e)$. $k > 0$ represents the

feedback gain. By implementing the heading rate controller as shown in Equation (24), the relative course $\chi$ of the UAV is indirectly controlled to follow the desired one $\chi_d$ generated by the LGVF. Therefore, in order to ensure that $\chi_d$ can be followed by the proposed controller, the allowable lower bound of $\omega_{max}$ must be determined. Before discussing the allowable lower bound of $\omega_{max}$, Lemma 1 and Lemma 2 are presented.

**Lemma 1:** *If the airspeed of the UAV is faster than the composition velocity, i.e., $v_s^2 \geq T_x^2 + T_y^2$, the following inequality is true.*

$$\frac{v_r}{\lambda_u(\psi)} \leq \frac{\left(v_s + \sqrt{T_x^2 + T_y^2}\right)^2}{v_s} \tag{25}$$

**Remark 3:** *The proof of Lemma 1 is seen in Appendix A. Lemma 1 can be used to determine the bound of $\dot{\chi}_d$ if the heading rate input constraint $\omega_{max}$ is provided. The relevant conclusions are shown in Lemma 2.*

**Lemma 2:** *If the heading rate control law given by Equation (24) is applied to the relative motion model of the UAV as shown in Equation (9), there exists a constant $k_1 : 0 < k_1 \leq \frac{k}{2}$, such that:* ① $\dot{\chi}_d \geq -\lambda_u(\psi)\omega_{max} + k_1\sin\chi_e, \chi_e \in [0, \pi)$; ② $\dot{\chi}_d \leq \lambda_u(\psi)\omega_{max} + k_1\sin\chi_e, \chi_e \in [-\pi, 0)$.

**Remark 4:** *The proof of Lemma 2 is seen in Appendix B. Lemma 2 provides the bound of the desired heading rate, i.e., $\left|\dot{\chi}_d - k_1\sin\chi_e\right| \leq \lambda_u(\psi)\omega_{max}, \chi_e \in [-\pi, \pi)$.*

Based on Lemmas 1 and 2, the allowable bound of $\omega_{max}$ is determined in Theorem 2.

**Theorem 2:** *In order to ensure that the UAV successfully performs standoff target tracking by implementing the heading rate controller as shown in Equation (24), and the input signal u never violates the heading rate constraint, i.e., $|u| \leq \omega_{max}$, the allowable lower bound of $\omega_{max}$ must satisfy the following condition.*

$$\omega_{max} \geq \omega_{inf} \triangleq \frac{4(v_s + \sqrt{T_x^2 + T_y^2})^2}{v_s r_d} \tag{26}$$

**Proof:** The proof is discussed in two cases in terms of the relative distance.

(a)   $r = 0$. According to the proposed heading rate control law as shown in Equation (24), when $r = 0$, $\chi_e = 0$. Now, the desired relative course rate is $\dot{\chi}_d = \frac{4v_r}{r_d}$. Then, according to Lemma 1, the following can be derived:

$$u = \left(\frac{4}{r_d}\right)\left(\frac{v_r}{\lambda_u(\psi)}\right) \leq \left(\frac{4}{r_d}\right)\left(\frac{\left(v_s + \sqrt{T_x^2 + T_y^2}\right)^2}{v_s}\right) \triangleq \omega_{inf} \tag{27}$$

It can be observed from Equation (27) that when $r = 0$, if $\omega_{max} \geq \omega_{inf}$, then $|u| \leq \omega_{max}$ is always true. This means that when $r = 0$, the proposed heading rate control satisfies the input constraint as shown in Equation (3).

(b)   $r > 0$. According to Lemma 2, the desired heading rate $\dot{\chi}_d$ is bounded, and the following inequalities can be derived:

$$\left| \dot{\chi}_d - k_1 sin\chi_e \right| \le \lambda_u(\psi)\omega_{max}$$
$$\Rightarrow \left| \frac{\dot{\chi}_d}{\lambda_u(\psi)} - 2k_1 sin\chi_e \right| \le \omega_{max}$$
$$\Rightarrow \left| \frac{\dot{\chi}_d}{\lambda_u(\psi)} - k sin\chi_e \right| \le \omega_{max} \tag{28}$$
$$\Rightarrow \left| \frac{\dot{\chi}_d}{\lambda_u(\psi)} - k \cdot \chi_e \right| \le \omega_{max}$$
$$\Rightarrow \left| u \right| \le \omega_{max}$$

It can be observed from Equation (28) that when $r > 0$, $|u| \le \omega_{max}$ is always true. Thus, the proof for Theorem 2 is completed. $\square$

**Remark 5:** *Theorem 2 provides the formulated inequality between the standoff radius $r_d$ and the maximum heading rate $\omega_{max}$ as follows:*

$$r_d \ge r_{d,min} \triangleq \frac{4\left( v_s + \sqrt{T_x^2 + T_y^2} \right)^2}{v_s \omega_{max}} \tag{29}$$

In other words, due to the input constraint $|u| \le \omega_{max}$, when the UAV performs the standoff target tracking mission with a constant airspeed $v_s$, the predefined standoff radius $r_d$ has an allowable lower bound. The minimum allowable standoff radius can be calculated according to Equation (29).

*3.3. Stability Analysis of Saturated Heading Rate Control Law*

Although Theorem 2 proves that the proposed heading rate control law satisfies the input constraint $|u| \le \omega_{max}$, whether $\chi$ converges to $\chi_d$ or not will be investigated in the following. In addition, if $\chi_e \to 0$ as $t \longrightarrow +\infty$, the convergence speed of $\chi_e$ also needs further discussion. Thus, Lemma 3 is presented.

**Lemma 3:** *If the UAV described by the relative motion model as shown in Equation (9) implements the heading rate control law given by Equation (24), there exists a positive constant $0 < k_2 \le \frac{k}{2} \le k_1$ such that ① $\dot{\chi}_e \ge -k_2 sin\chi_e > 0$, $\chi_e \in [-\pi, 0)$; ② $\dot{\chi}_e \le -k_2 sin\chi_e < 0$, $\chi_e \in [0, \pi)$.*

**Remark 6:** *The proof of Lemma 3 is seen in Appendix C. According to Lemma 3, it can be derived that*

$$\chi_e \dot{\chi}_e \le -k_2 \chi_e sin\chi_e \le 0 \tag{30}$$

Equation (30) shows that for every initial relative course error $\chi_e(t_0) \in [-\pi, \pi)$, the relative course error $\chi_e \to 0$ as $t \to +\infty$. Thus, the convergence of the proposed heading rate control law is proved in Lemma 3. In addition, Lemma 3 also gives a lower bound on the convergence rate $\dot{\chi}_e$, which is defined as follows:

$$\left| \dot{\chi}_e \right| \ge \left| \dot{\chi}_e \right|_{inf} = k_2 |sin\chi_e|, \; \chi_e \in [-\pi, \pi) \tag{31}$$

By solving the differential inequality shown in (31), the relative course error $\chi_e$ varies according to the following inequality:

$$\left| tan\left( \frac{\chi_e}{2} \right) \right| \le \left| tan\left( \frac{\chi_e(t_0)}{2} \right) \right| e^{-k_2 t} \tag{32}$$

Equation (32) can be used to analyze the bound of the relative distance $r$ between the UAV and the target. The relevant conclusions are shown in Lemma 4.

**Lemma 4:** *Given an arbitrary initial relative course error $\chi_e(t_0) \in [-\pi, \pi)$, if the UAV described by the relative motion model as shown in Equation (9) implements the heading rate control law given by Equation (24), the relative distance $r$ has an upper bound $r_{sup}$ as follows:*

$$r_{sup} = r_0 + \frac{v_s + T}{k_2} ln \left| \frac{tan\left(\frac{\chi_{e0}}{2}\right)}{tan\left(\frac{\alpha_0}{2} - \frac{\pi}{4}\right)} \right| \tag{33}$$

where $T = \sqrt{T_x^2 + T_y^2}$ and $\alpha_0 = arctan\left(\frac{r_0}{r_d}\right)$. The proof of Lemma 4 is seen in Appendix D.

Based on Lemma 3 and Lemma 4, Theorem 3 presents a complete proof for the stability of the proposed heading rate control law with control input constrains.

**Theorem 3:** *Given an arbitrary initial relative course error $\chi_{e0} \in [-\pi, \pi)$, if the UAV described by the relative motion model as shown in Equation (9) implements the heading rate control law given by Equation (24), the trajectory of the UAV asymptotically converges to the standoff circle, i.e., $r \longrightarrow r_d$ and $\chi \rightarrow \chi_d$ as $t \longrightarrow +\infty$.*

**Proof:** Let $\mu > 0$, and a Lyapunov function be introduced as follows:

$$V = \frac{1}{2}(r - r_d)^2 + \frac{1}{2}\mu\chi_e^2 \tag{34}$$

Differentiating Equation (34) with respect to time $t$, the outcome is

$$\dot{V} = (r - r_d)\dot{r} + \mu\chi_e\dot{\chi}_e \tag{35}$$

(a) If $|\chi_e| \in \left[\frac{\pi}{2}, \pi\right]$, according to Lemma 4, we obtain $(r - r_d)\dot{r} = (r - r_d)v_r cos(\chi - \theta)$ $\leq (r_{sup} + r_d)(v_s + T)$. And according to Lemma 3, we have $\mu\chi_e\dot{\chi}_e \leq \mu\chi_e(-k_2 sin\chi_e)$ $\leq \mu\left(\frac{\pi}{2}\right)(-k_2 sin\chi_{e0})$. Together, they yield that

$$\dot{V} \leq (r_{sup} + r_d)(v_s + T) - \mu k_2 \left(\frac{\pi}{2}\right) sin\chi_{e0} \tag{36}$$

It can be observed that from Equation (36), if $\mu \geq \mu_1 \triangleq \frac{2(r_{sup}+r_d)(v_s+T)}{k_2 \pi sin\chi_{e0}}$, then $\dot{V} \leq 0$.

(b) If $|\chi_e| \in \left[0, \frac{\pi}{2}\right]$, according to Lemma 3, one obtains

$$\mu\chi_e\dot{\chi}_e \leq -\mu k_2\chi_e sin\chi_e \tag{37}$$

According to the relative motion model of the UAV as shown in Equation (9), we have

$$\dot{r} = -\left(\frac{r^2 - r_d^2}{r^2 + r_d^2}\right)v_r cos\chi_e - v_r sin\phi_{GF} sin\chi_e \tag{38}$$

Substituting (37) and (38) into (35) yields

$$\dot{V} \leq -\frac{(r - r_d)^2 (r + r_d)v_r}{r^2 + r_d^2} cos\chi_e - v_r(r - r_d) sin\phi_{GF} sin\chi_e - \mu k_2 \chi_e sin\chi_e \tag{39}$$

For the second term in Equation (39), it holds that

$$-v_r(r - r_d) sin\phi_{GF} sin\chi_e \leq 2v_r |r - r_d| \left| sin\frac{\chi_e}{2} \right| \tag{40}$$

For the third term in Equation (39), it holds that

$$\chi_e sin\chi_e \geq 2sin^2\left(\frac{\chi_e}{2}\right) \tag{41}$$

Substituting Equations (40) and (41) into Equation (39) yields

$$\dot{V} \le -\frac{(r-r_d)^2(r+r_d)v_r}{r^2+r_d^2} + 2v_r|r-r_d|\left|sin\frac{\chi_e}{2}\right| - 2\mu k_2 sin^2\left(\frac{\chi_e}{2}\right) \tag{42}$$

Solving Equation (42) by completing the square, it is easy to know that if $\mu \ge \mu_2 \triangleq \frac{(r_{sup}^2+r_d^2)}{k_2(r_{sup}+r_d)}$, then

$$\dot{V} \le -2\mu k_2 sin^2\left(\frac{\chi_e}{2}\right) - \left\{\left[\frac{(r+r_d)}{r^2+r_d^2}\right]^{\frac{1}{2}}|r-r_d| - \left[\frac{(r^2+r_d^2)}{r+r_d}\right]^{\frac{1}{2}}\left|sin\left(\frac{\chi_e}{2}\right)\right|\right\}^2 \le 0 \tag{43}$$

Therefore, when $\mu \ge max\{\mu_1, \mu_2\}$, the time derivative of the Lyapunov function is always non-positive, i.e., $\dot{V} \le 0$. It can be observed from Equation (43) that $\dot{V} = 0$ implies that $r = r_d$ and $\chi_e = 0$. According to LaSalle's invariance principle, it can be concluded that $r \longrightarrow r_d$ and $\chi \to \chi_d$ as $t \longrightarrow +\infty$. This means that the trajectory of the UAV asymptotically converges to the standoff circle. Thus, the proof of Theorem 3 is completed. □

**Remark 7:** *According to Theorem 3, it is concluded that by implementing the proposed heading rate control law, the relative distance and the relative course of the UAV converge to the desired values. This means that the control objectives 1 and 2 proposed in Section 2.2 are achieved, and the UAV tracks the moving target while maintaining the desired standoff distance successfully.*

## 4. Cooperative Standoff Tracking Using Multiple UAVs

When a team of UAVs is used to track a ground-based moving target, coordination between aircraft is necessary to avoid collisions and to maximize sensor coverage of the target. A possible solution to this coordination problem is the so-called "phase separation" approach whereby the UAVs fly along the standoff circle with an equal intervehicle phase separation angle with respect to their neighbors.

Ref. [25] proposes a space phase separation algorithm (SPSA) to achieve the desired angular spacing, as illustrated in Figure 3. The UAVs are distributed counterclockwise in the standoff circle according to the ascending sequence of their unique identity numbers $A_i (i = 1, 2, \dots N)$. Each UAV $A_i$ independently calculates its airspeed control input $v_{s,i}$ based on its own phase angle $\theta_i$, the phase angle $\theta_{left} = \theta_{i-1}$ of its left neighbor $A_{i-1}$, and the phase angle $\theta_{right} = \theta_{i+1}$ of its right neighbor $A_{i+1}$. For $A_1$, its left neighbor is $A_N$ and its right neighbor is $A_2$. For $A_N$, its left neighbor is $A_{N-1}$ and its right neighbor is $A_1$.

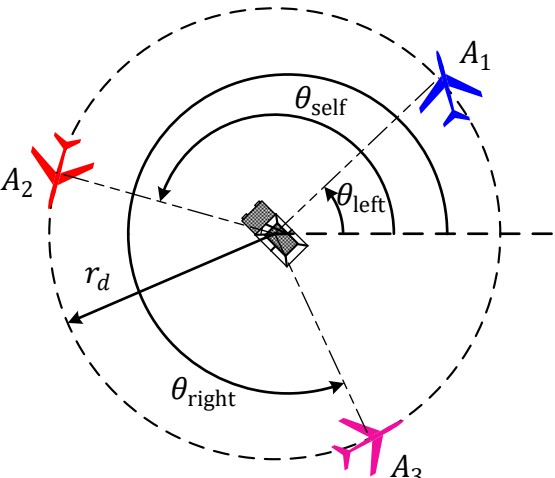

**Figure 3.** Illustration of phase separation control for UAVs.

In [25], the space phase separation angle of $A_i$, $\Delta\theta_i \in [-\pi, \pi)$, is defined as $\Delta\theta_i = \theta_i - \theta_{left}$. Similarly, the phase separation angle of the right neighbor $A_{i+1}$ is defined as $\Delta\theta_{right} = \theta_{right} - \theta_i$. The space phase separation error of $A_i$, $e_{\theta,i} \in [-\pi, \pi)$ is defined as $e_{\theta,i} = \Delta\theta_i - \theta_d$. Similarly, the space phase separation error of the right neighbor $A_{i+1}$ is defined as $e_{\theta,right} = \Delta\theta_{right} - \theta_d$. Thus, the airspeed control input of $A_i$ can be designed as $v_{s,i} = k_\theta(e_{\theta,right} - e_{\theta,i})r_i + v_{sd}$. Where $k_\theta > 0$, $r_i$ represents the relative distance between $A_i$ and the target. $v_{sd}$ represents the desired airspeed when the team of UAVs hover around the target along a standoff circle synchronously. In this paper, $v_{sd}$ is called the standoff speed for short.

However, the SPSA method proposed in [25] has two disadvantages. (i) The airspeed limitation, as shown in Equation (2), is not considered in SPSA; (ii) the space phase separation $\Delta\theta_i$ and corresponding error $e_{\theta,i}$ are between $[-\pi, \pi)$, and thus they are discontinuous. This discontinuity will lead to oscillations in the airspeed control input, resulting in poor tracking performance of the UAVs. This shortcoming has been confirmed by simulation results shown in Figure 9 in [25]. In order to avoid the discontinuity of the space phase angle, the space phase is replaced by a new notion called the temporal phase, which can also be used to represent the distribution of the UAVs on the standoff circle. Based on the temporal phase, a new temporal phase separation algorithm (TPSA) is proposed to achieve the desired temporal phase separation in a cooperative standoff target tracking mission.

### 4.1. Airspeed Controller Design for Temporal Phase Separation

In the SPSA method, the angle $\theta_i$ is utilized to represent the space phase of the UAV $A_i$. The space phase describes the distribution of the UAVs which remain in a circle around the target. Similarly, we can also introduce the temporal phase to equivalently describe the distribution of the UAVs. The temporal phase of $A_i$ is defined as follows:

$$\tau_i(\theta_i) = \begin{cases} \frac{2\pi}{T_\tau}\left[\int_0^{\theta_i + 2\pi}\left(\frac{r_d}{v_{rd}}\right)d\theta\right] - \pi, & if \ \theta_i \in [-\pi, 0) \\ \frac{2\pi}{T_\tau}\left[\int_0^{\theta_i}\left(\frac{r_d}{v_{rd}}\right)d\theta\right] - \pi & if \ \theta_i \in [0, \pi) \end{cases} \tag{44}$$

where $\tau_i(\theta_i) \in [-\pi, \pi)$ represents the temporal phase of $A_i$. $v_{rd}$ is the desired relative speed of $A_i$ with respect to the target and can be expressed as follows.

$$v_{rd} = \left[v_{sd}^2 + T_x^2 + T_y^2 - 2v_{sd}\left(T_x cos\psi_d + T_y sin\psi_d\right)\right]^{\frac{1}{2}} \tag{45}$$

where $\psi_d$ represents the desired heading when $A_i$ flies along the standoff circle.

$$\psi_d = arcsin\left(\frac{T_y cos\chi_d - T_x sin\chi_d}{v_{sd}}\right) + \chi_d \tag{46}$$

In Equation (44), $T_\tau$ represents the time required for the UAV to complete a circle of flight along the standoff circle with the desired airspeed $v_{sd}$. $T_\tau$ is used to normalize the flight time of the UAV and is defined as $T_\tau = \int_0^{2\pi}\left(\frac{r_d}{v_{rd}}\right)d\theta$. It can be observed from Equation (44) that the temporal phase $\tau_i(\theta_i)$ represents the normalized time required for the UAV $A_i$ to fly from space phase angle 0 to the current space phase angle $\theta_i$ along the standoff circle.

Therefore, the temporal phase separation between $A_i$ and its neighbor is defined by the difference in their temporal phases.

$$\tau_{i-1,i} = \tau_{i-1} - \tau_i \tag{47}$$

Suppose there are $N(N \geq 2)$ UAVs, a leader–follower formation strategy is used in this paper. Without loss of generality, it assumes that $A_1$ is the leader and its airspeed is the desired one, e.g., $v_{s,1} = v_{sd}$, and is held constant. Then, $A_2$ follows $A_1$, and adjusts its airspeed to achieve the desired temporal separation with its leader $A_1$. Similarly, $A_i$ follows

its leader $A_{i-1}$ and achieves the desired temporal separation by varying its airspeed. The temporal separation error is defined as follows:

$$\Delta\tau_i = \tau_{i-1} - \tau_i - \tau_d \tag{48}$$

where $\tau_d$ represents the desired separation. The airspeed input is designed as follows:

$$\begin{cases} v_{s,1} = v_{sd} \\ v_{s,i} = v_{sd} + \left(\frac{\Delta v \cdot \Delta\tau_i}{\pi}\right)\left(\frac{r_{i-1}^2+r_d^2}{r_{i-1}^2+r_i^2}\right), i > 1 \end{cases} \tag{49}$$

where $\Delta v > 0$ represents the airspeed increment of the UAV in one time step, and it reflects the performance of the onboard autopilot.

Before verifying that the proposed controller Equation (49) always satisfies the minimum and maximum airspeed constraints in Equation (2), Assumption 5 is introduced as follows:

**Assumption 5:** *There exist appropriate values of $v_{sd}$ and $\Delta v$, so that the following constraints are satisfied, $v_{sd} - \Delta v > \sqrt{T_x^2 + T_y^2}$ and $v_{min} + \Delta v \leq v_{sd} \leq v_{max} - \Delta v$.*

**Remark 8:** *Generally speaking, the airspeed of the UAV is always faster than the target speed and the wind speed, thus the constraint $v_{sd} - \Delta v > \sqrt{T_x^2 + T_y^2}$ could be satisfied. In addition, $\Delta v$ is dependent on the performance of the autopilot, which definitely satisfies the requirement of $v_{min} + \Delta v \leq v_{sd} \leq v_{max} - \Delta v$. Therefore, Assumption 5 is reasonable. Based on Assumption 5, the following Theorem 4 is proposed.*

**Theorem 4:** *If the standoff speed $v_{sd}$ and the speed increment $\Delta v$ satisfy the inequality conditions presented in Assumption 5, then the proposed controller given by Equation (49) always satisfies the minimum and maximum airspeed constraints given by Equation (2).*

**Proof:** According to Equation (49), let $\gamma = v_{s,i} - v_{sd} = \Delta v \cdot \alpha \cdot \beta$, where $\alpha = \frac{\Delta\tau_i}{\pi}$ and $\beta = \frac{r_{i-1}^2+r_d^2}{r_{i-1}^2+r_i^2}$. Due to $\Delta\tau_i \in [-\pi, \pi)$, then $|\Delta\tau_i| \leq \pi$ and $|\alpha| \leq 1$. If $r_i \geq r_d$, then $0 < \beta \leq 1$, i.e., $|\beta| \leq 1$.

$$|\gamma| = \Delta v \cdot |\alpha| \cdot |\beta| \leq \Delta v \tag{50}$$

Considering $v_{s,i} = v_{sd} + \gamma$, yields

$$v_{sd} - |\gamma| \leq (|v_{s,i}| = |v_{sd} + \gamma|) \leq v_{sd} + |\gamma| \tag{51}$$

Substituting (50) into (51) yields

$$v_{sd} - \Delta v \leq |v_{s,i}| \leq v_{sd} + \Delta v \tag{52}$$

According to Assumption 5, it is obtained that $v_{min} \leq v_{sd} - \Delta v \leq |v_{s,i}| \leq v_{sd} + \Delta v \leq v_{max}$, which implies that $0 < v_{min} \leq v_{s,i} \leq v_{max}$. Therefore, it can be concluded that the proposed airspeed control law given by Equation (49) always satisfies the minimum and maximum airspeed constraints given by Equation (2). Thus, the proof for Theorem 3 is completed. □

*4.2. Stability Analysis of Airspeed Control Law*

A stability analysis of the airspeed control law is provided in the following Theorem 5.

**Theorem 5:** *For the relative motion model of a UAV described in Equation (9), if the heading rate control law given by Equation (24) and the cooperative airspeed control laws given by Equation (49) are applied to a team of UAVs, then each aircraft $A_i$ can achieve equal temporal separation and fly along the standoff circle, i.e., as $t \longrightarrow +\infty, \Delta\tau_i \longrightarrow 0, i = 2, 3, \ldots, N$.*

**Proof:** Differentiating Equation (48) with respect to time $t$ yields

$$\Delta\dot{\tau}_i = \dot{\tau}_{i-1} - \dot{\tau}_i = \left(\frac{2r_d\pi}{T_\tau}\right)\left[\left(\frac{\dot{\theta}_{i-1}}{v_{rd,(i-1)}}\right) - \left(\frac{\dot{\theta}_i}{v_{rd,i}}\right)\right] \qquad (53)$$

According to the relative motion model given by Equation (9), one obtains

$$\dot{\theta}_i = \frac{v_{r,i}}{r_i}sin\phi_{GF,i} = \frac{v_{r,i}}{r_i}\cdot\frac{2r_ir_d}{r_i^2 + r_d^2} \qquad (54)$$

Suppose that aircraft $A_i$ ($i = 2, 3, \ldots, N$). implementing the proposed heading rate control law given by Equation (24), has been converged on the standoff circle, i.e., $r_i = r_d$. Therefore,

$$\Delta\dot{\tau}_i = \left(-\frac{2\pi}{T_\tau}\right)\left[\frac{v_{r,i} - v_{rd,i}}{v_{rd,i}} - \frac{v_{r,(i-1)} - v_{rd,(i-1)}}{v_{rd,(i-1)}}\right] \qquad (55)$$

Let $\sigma = T_x cos\psi_{rd,i} + T_y sin\psi_{rd,i}$, we obtain

$$v_{r,i}^2 - v_{rd,i}^2 = (v_{s,i} - v_{sd})(v_{s,i} + v_{sd} - 2\sigma) \qquad (56)$$

According to the cooperative airspeed control laws given by Equation (49), one has

$$v_{s,i} - v_{sd} = \left(\frac{\Delta v\cdot\Delta\tau_i}{\pi}\right) \qquad (57)$$

Substituting Equation (57) into Equation (56) yields

$$v_{r,i} - v_{rd,i} = K_i\Delta\tau_i \qquad (58)$$

where $K_i \triangleq \frac{\Delta v\left(v_{s,i} + v_{sd} - 2\sigma\right)}{\pi\left(v_{r,i} + v_{rd,i}\right)}$. Substituting Equation (58) into Equation (55) and setting $v_{r,1} = v_{rd,1}$ yields

$$\begin{cases} \Delta\dot{\tau}_2 = -\frac{2\pi}{T_\tau}\left(\frac{K_2\Delta\tau_2}{v_{rd,2}} - 0\right) \\ \Delta\dot{\tau}_3 = -\frac{2\pi}{T_\tau}\left(\frac{K_3\Delta\tau_3}{v_{rd,3}} - \frac{K_2\Delta\tau_2}{v_{rd,2}}\right) \\ \vdots \\ \Delta\dot{\tau}_i = -\frac{2\pi}{T_\tau}\left(\frac{K_i\Delta\tau_i}{v_{rd,i}} - \frac{K_{(i-1)}\Delta\tau_{(i-1)}}{v_{rd,(i-1)}}\right) \end{cases} \qquad (59)$$

Let $\Delta\tau = (\Delta\tau_2, \Delta\tau_3, \ldots, \Delta\tau_N)^T$ and $\varepsilon_i = \frac{2\pi K_i}{T_\tau v_{rd,i}}$, one derives

$$\Delta\dot{\tau} = -\begin{bmatrix} \varepsilon_2 & 0 & \cdots & \cdots \\ -\varepsilon_2 & \varepsilon_3 & 0 & \cdots \\ \vdots & -\varepsilon_3 & \ddots & 0 \\ \vdots & \vdots & -\varepsilon_{N-1} & \varepsilon_N \end{bmatrix}\Delta\tau \qquad (60)$$

It is easy to prove that the cascade connected system shown in Equation (60) is asymptotically stable, i.e., $\Delta\tau \longrightarrow 0$ as $t \longrightarrow +\infty$. Thus, the proof for Theorem 5 is completed. □

## 5. Estimator for the Composition Velocity of Background Wind and Target's Motion

In the previous section, the proofs and analyses are based on the assumption that the composition velocity is accurately known a priori. However, in practical applications, this assumption is not always true. Therefore, in this section, an estimator is proposed to estimate the composition velocity in real time.

*5.1. Principle of the Composition Velocity Estimator*

Suppose that the composition velocity $T = (T_x, T_y)$ is bounded, and its upper bound is known a priori, denoted as $T^* = sup_{t \geq 0}\{\|T\|_2\}$. According to [31], the target motion and wind are regarded as an external disturbance. And the composition velocity of the external disturbance can be expressed in the form of a hyperbolic tangent function as follows:

$$T_x = T^* tanh\varphi_x; T_y = T^* tanh\varphi_y \tag{61}$$

The parameters of the external disturbance, $\varphi_x$ and $\varphi_y$, are regarded as estimated variables to obtain the estimated results of the composition velocity. $\hat{T} = (\hat{T}_x, \hat{T}_y)$ is introduced to denote the estimation of $(T_x, T_y)$, then it also can be expressed in the form of a hyperbolic tangent function as follows:

$$\hat{T}_x = T^* tanh\hat{\varphi}_x; \hat{T}_y = T^* tanh\hat{\varphi}_y \tag{62}$$

where $(\hat{\varphi}_x, \hat{\varphi}_y)$ is an estimated result of the external disturbance parameters. Thus, our goal is to design an estimator which can guarantee that the estimated disturbance parameters can asymptotically coverage to the true values, i.e., $(\hat{\varphi}_x, \hat{\varphi}_y) \rightarrow (\varphi_x, \varphi_y)$ as $t \longrightarrow +\infty$.

Therefore, the dynamics of the estimated relative position are formulated as follows:

$$\dot{\hat{x}}_r = v_s cos\psi - \hat{T}_x + k_3\tilde{x}_r; \dot{\hat{y}}_r = v_s sin\psi - \hat{T}_y + k_3\tilde{y}_r \tag{63}$$

where $k_3 > 0$ is a positive constant. $(\tilde{x}_r, \tilde{y}_r)$ is the estimated error in the relative position.

$$\tilde{x}_r = x_r - \hat{x}_r; \tilde{y}_r = y_r - \hat{y}_r \tag{64}$$

where $(\hat{x}_r, \hat{y}_r)$ represents the estimated relative position. Then, differentiating Equation (64) with respect to time $t$, and according to Equation (63), the dynamics of the relative position estimate error can be expressed as follows:

$$\dot{\tilde{x}}_r = -\tilde{T}_x - k_3\tilde{x}_r; \dot{\tilde{y}}_r = -\tilde{T}_y - k_3\tilde{y}_r \tag{65}$$

where $(\tilde{T}_x, \tilde{T}_y)$ represents the estimated error in the composition velocity:

$$\tilde{T}_x = T_x - \hat{T}_x; \tilde{T}_y = T_y - \hat{T}_y \tag{66}$$

It is worth noting that the errors $(\tilde{T}_x, \tilde{T}_y)$ could be reduced by $(\hat{\varphi}_x, \hat{\varphi}_y)$. Therefore, let $k_4 > 0$ be a positive constant, $(\hat{\varphi}_x, \hat{\varphi}_y)$ is updated as follows:

$$\dot{\hat{\varphi}}_x = -k_4\tilde{x}_r; \dot{\hat{\varphi}}_y = -k_4\tilde{y}_r \tag{67}$$

The pseudocode of the composition velocity estimation algorithm (CVEA) is illustrated in Algorithm 1.

It is worth explaining that the differences between the CVEA proposed in this paper and in [31] are shown in Figure 4. Point $A$ represents the current position of the UAV; the corresponding azimuth angle is denoted by $\theta$. Suppose $P$ is the point with the corresponding azimuth angle $\theta$ on the desired obit of the LGVF, and the position of $P$ can be calculated according to Equation (16). Generally speaking, it is hoped that the two points $A$ and $P$ should coincide. However, due to the effect of external disturbances, there exists an offset between the actual vehicle trajectory and the desired LGVF orbit, denoted as $\tilde{r}(\theta) = \overrightarrow{AP} = (\tilde{x}_r, \tilde{y}_r)$. The offset can be used to estimate the composition velocity. In this paper, the CVEA approach uses the whole offset vector $[\tilde{x}_r, \tilde{y}_r]^T$ to obtain the estimation of the composition velocity. However, Ref. [31] only uses the radial distance of the offset, i.e., $\tilde{r}(\theta) = \sqrt{\tilde{x}_r^2 + \tilde{y}_r^2}$, in the estimator. Theoretically speaking, because the feedback item contains more information compared to what was used in [31], the composition velocity can be estimated faster and more accurately in the proposed CVEA.

**Algorithm 1** Pseudocode of the composition velocity estimation algorithm (CVEA)

**Input:** The estimate results $\left[\hat{T}_x^{\ominus}, \hat{T}_y^{\ominus}, \hat{\varphi}_x^{\ominus}, \hat{\varphi}_y^{\ominus}, \hat{x}_r^{\ominus}, \hat{y}_r^{\ominus}\right]^T$ at time $t_{k-1}$, and the measured relative position $[x_r, y_r]^T$ at the present $t_k$.

**Output:** The estimate results $\left[\hat{T}_x^{\oplus}, \hat{T}_y^{\oplus}, \hat{\varphi}_x^{\oplus}, \hat{\varphi}_y^{\oplus}, \hat{x}_r^{\oplus}, \hat{y}_r^{\oplus}\right]^T$ at time $t_k$.

1. Update the estimate error in the relative position as follows

$$\tilde{x}_r = x_r - \hat{x}_r^{\ominus}; \tilde{y}_r = y_r - \hat{y}_r^{\ominus}$$

where $(x_r, y_r)$ can be regarded as the measurement update at time $t_k$; $(\hat{x}_r^{\ominus}, \hat{y}_r^{\ominus})$ represent the prior estimation of the relative position at time $t_{k-1}$.

2. Calculate a posterior estimate of the parameters $(\varphi_x, \varphi_y)$ as follows

$$\hat{\varphi}_x^{\oplus} = \hat{\varphi}_x^{\ominus} - k_4 \tilde{x}_r \cdot \Delta t; \hat{\varphi}_y^{\oplus} = \hat{\varphi}_y^{\ominus} - k_4 \tilde{y}_r \cdot \Delta t$$

where $\Delta t = t_k - t_{k-1}$ represents the time step.

3. A posteriori estimation of the composition velocity $(T_x, T_y)$ is calculated as follows

$$\hat{T}_x^{\oplus} = T^* tanh \hat{\varphi}_x^{\oplus}; \hat{T}_y^{\oplus} = T^* tanh \hat{\varphi}_y^{\oplus}$$

4. Update the posterior estimate of the relative position as follows

$$\hat{x}_r^{\oplus} = \hat{x}_r^{\ominus} + (v_{sd} cos\psi - \hat{T}_x^{\oplus} + k_3 \tilde{x}_r)\Delta t; \hat{y}_r^{\oplus} = \hat{y}_r^{\ominus} + (v_{sd} sin\psi - \hat{T}_y^{\oplus} + k_3 \tilde{y}_r)\Delta t$$

5. Update time $t_{k-1} \leftarrow t_k$, go to Step 1 to start the next round of estimation.

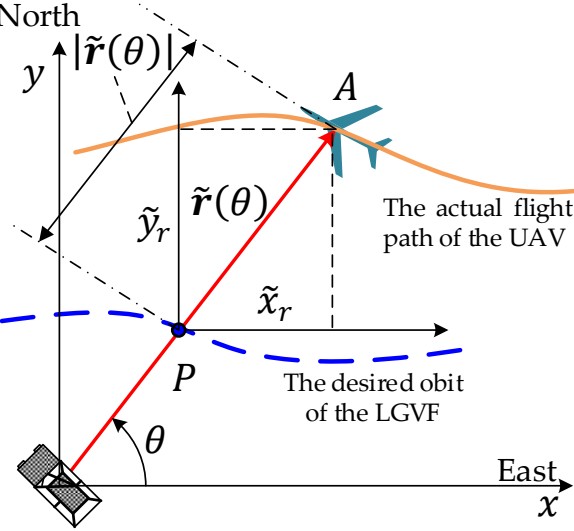

**Figure 4.** Principle of CVEA.

*5.2. Convergence Analysis of the Composition Velocity Estimator*

A convergence analysis of the composition velocity estimator is provided in the following Theorem 6.

**Theorem 6:** *The composition velocity estimator given by Equations (63)–(67) is asymptotically stable, i.e., $\hat{T}_x \rightarrow T_x$ and $\hat{T}_y \rightarrow T_y$ as $t \longrightarrow +\infty$.*

**Proof:** Consider the following Lyapunov function:

$$V = \frac{1}{2}\tilde{x}_r^2 + \frac{1}{2}\tilde{y}_r^2 \quad + \frac{T^*}{k_4}[ln(cosh(\hat{\varphi}_x)) - \hat{\varphi}_x \cdot tanh(\varphi_x)] \\ + \frac{T^*}{k_4}[ln(cosh(\hat{\varphi}_y)) - \hat{\varphi}_y \cdot tanh(\varphi_y)] \tag{68}$$

(a)    Firstly, we prove that the Lyapunov function $V$ is negative semi-definite, i.e., $\dot{V} \leq 0$.

Differentiating Equation (68) with respect to time $t$, and according to Equations (63)–(67) yields

$$\dot{V} = \tilde{x}_r \dot{\tilde{x}}_r + \tilde{y}_r \dot{\tilde{y}}_r - \frac{\dot{\hat{\varphi}}_x}{k_4} \tilde{T}_x - \frac{\dot{\hat{\varphi}}_y}{k_4} \tilde{T}_y = -k_3 \left( \tilde{x}_r^2 + \tilde{y}_r^2 \right) \leq 0 \tag{69}$$

(b)　Secondly, it is shown that the Lyapunov function $V$ is lower bounded, i.e., $V \geq V_{inf}$.

An assistant function is defined as $H(x) = ln(cosh(x)) - x \cdot tanh(x_0)$. Differentiating $H(x)$ with respect to $x$, we obtain $\frac{\partial H}{\partial x} = tanh(x) - tanh(x_0)$. This means that when $x = x_0$, $H(x)$ reaches the minimum, i.e., $H(x) \geq H(x_0) \geq -ln2$. Thus, when $\hat{\varphi}_x = \varphi_x$ and $\hat{\varphi}_y = \varphi_y$, the Lyapunov function $V$ reaches the following lower bound $V \geq V_{inf} \triangleq -\frac{2T^*}{k_4} ln2$. Therefore, the function $V$ is lower bounded.

(c)　Finally, we prove that the function $\ddot{V}$ is bounded.

Differentiating Equation (69) with respect to time $t$, one obtains

$$\ddot{V} = -2k_3 \left( \tilde{x}_r \cdot \dot{\tilde{x}}_r + \tilde{y}_r \cdot \dot{\tilde{y}}_r \right) = 2k_3 \left[ \tilde{x}_r \left( \tilde{T}_x + k_3 \tilde{x}_r \right) + \tilde{y}_r \left( \tilde{T}_y + k_3 \tilde{y}_r \right) \right] \tag{70}$$

Because $\tilde{x}_r, \tilde{y}_r, \tilde{T}_x, \tilde{T}_y$ are all bounded in the proposed estimator, thus the function $\ddot{V}$ is also bounded. Therefore, according to Barbalat's lemma, it is concluded that $\dot{V} \to 0$ as $t \longrightarrow +\infty$. This implies that $\tilde{x}_r \to 0$ and $\tilde{y}_r \to 0$ as $t \longrightarrow +\infty$. In addition, it can be proved that $\ddot{\tilde{x}}_r$ is also bounded, and thus we can conclude that $\dot{\tilde{x}}_r = -\tilde{T}_x - k_3 \tilde{x}_r \to 0$ according to Barbalat's lemma. In other words, $\tilde{T}_x \to 0$. Similarly, it can also be proved that $\tilde{T}_y \to 0$. We can summarize that $\hat{T}_x \to T_x$ and $\hat{T}_y \to T_y$ as $t \longrightarrow +\infty$. Thus, the proof for Theorem 6 is completed. □

### 5.3. Limitation of the Composition Velocity Estimator

A signal flow diagram of the CVEA is shown in Figure 5. The true value of the target's position should be known when the CVEA is implemented. However, the accurate target position cannot be obtained. Thus, in order to estimate the composition velocity of the unknown wind and target's motion, Assumption 2 in Section 2 is required. The application of the proposed CVEA is limited by Assumption 2. The target's states (e.g., position, velocity, and acceleration) can be estimated using various Kalman filter algorithms, and the target's motion and wind velocity can be obtained separately. In this paper, we keep the target's location out of the scope since we aim at providing a solid formulation concerning the problem of coordinated standoff tracking of a ground target. In future work, a target state estimate algorithm will be designed and integrated into the framework of the CVEA.

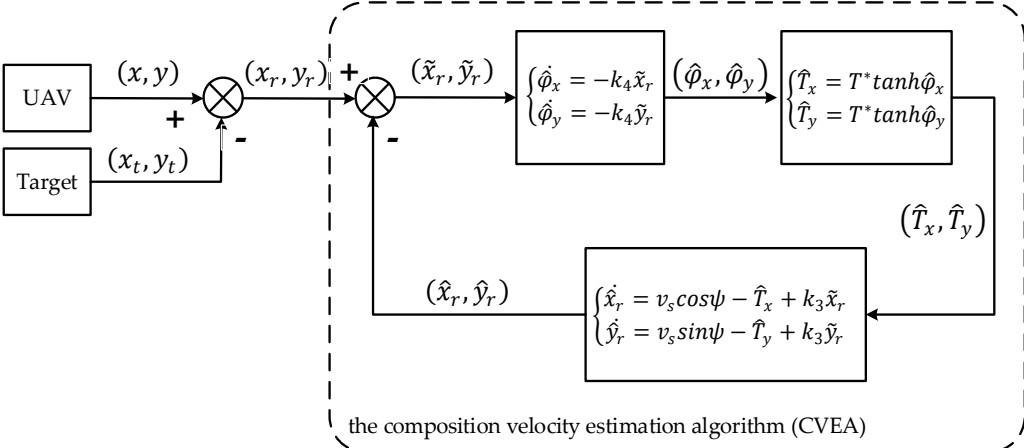

**Figure 5.** Signal flow diagram of CVEA.

## 6. Decentralized Control and Coordination Architecture of Standoff Target Tracking

A more detailed control and coordination architecture of the proposed coordinated standoff target tracking algorithm (CSTTA) is shown in Figure 6. The proposed CSTTA is scalable and does not require significant computation and communication power.

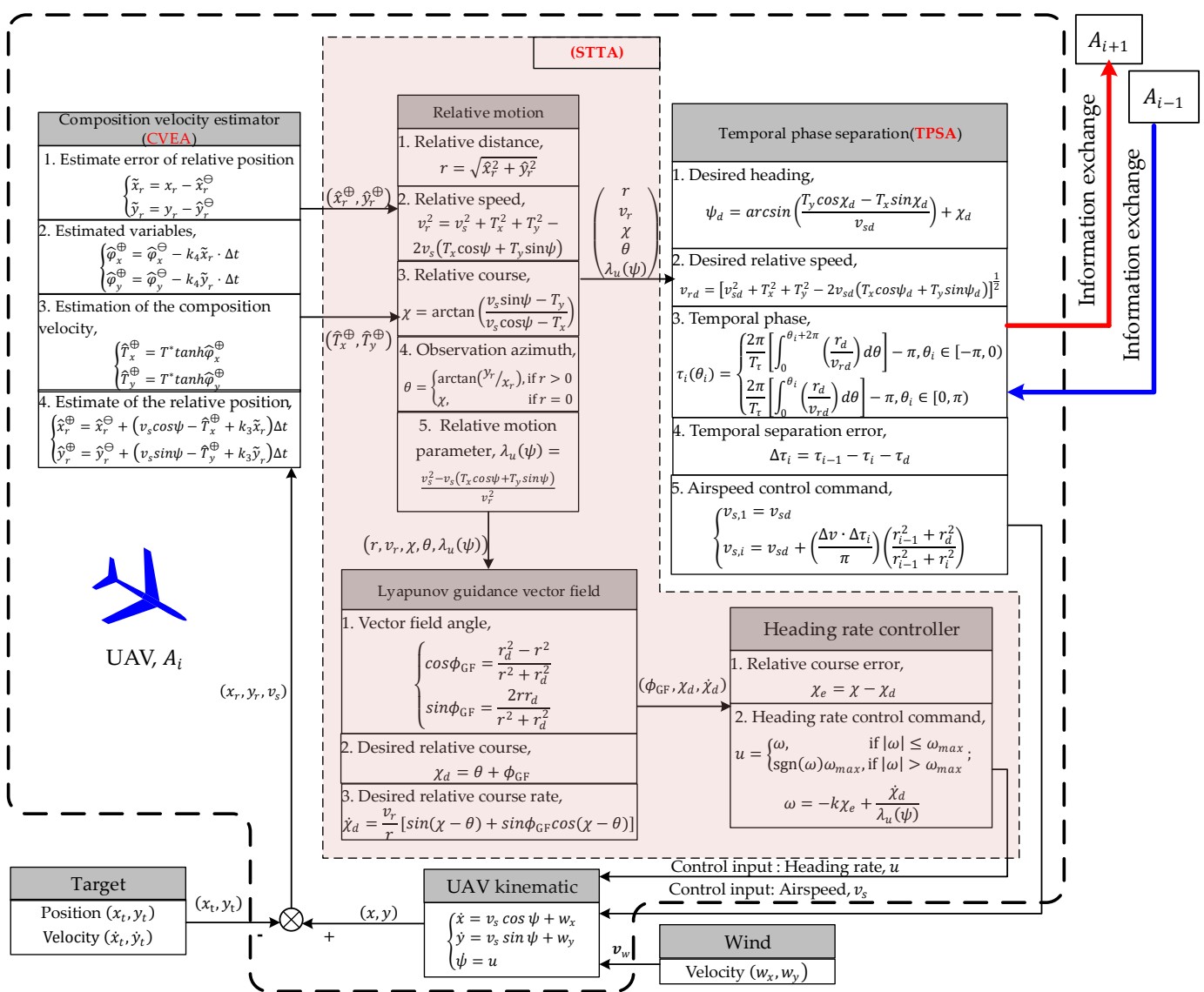

**Figure 6.** Decentralized control and coordination architecture of the proposed coordinated standoff target tracking algorithm (CSTTA).

On one hand, it can be observed from Figure 6 that all variables or signals are given their analytic solutions in the CSTTA. This is different from the MPC method proposed in Ref. [16]; the CSTTA proposed in this paper does not require an iterative optimization process, so it has a lower computational complexity. Therefore, the proposed CSTTA can be used in multi-UAV collaborative applications with high real-time requirements. It is worth noting that the temporal phase can be obtained by implementing Romberg integration, which has low computational complexity and will not increase the computational burden of the whole system.

On the other hand, it can be observed from Figure 6 that the control and coordination architecture is decentralized. In a distributed fashion, each UAV only transmits messages to its neighbors that are in its communication range. In the proposed CSTTA, $A_i$ computes its own temporal separation error $\Delta\tau_i$ by exchanging information with its leader $A_{i-1}$. And

then $A_i$ generates its airspeed command $v_{s,i}$ and exchanges this command with its follower $A_{i+1}$. Therefore, the proposed CSTTA is a feasible approach to reduce the communication between UAVs.

It is worth noting that in our paper it is assumed that the communication between UAVs is perfect, without any restrictions. However, in the real-world, the wireless communication network between UAVs is vulnerable to errors and time delays, which may lead to performance degradation or even instability. In future work, we will analyze the effects of the potential communication constraints, which is a critical issue for the successful operation of multiple UAVs.

As shown in Figure 6, the proposed CSTTA method includes three parts, as follows: (i) the saturated heading rate controller, which is the standoff target tracking algorithm (STTA) for one single UAV; (ii) the airspeed controller based on the temporal phase separation algorithm (TPSA), which is used to achieve the desired temporal phase separation in cooperative standoff target tracking with multiple UAVs; (iii) the composition velocity estimation algorithm (CVEA), which is used to ensure the stability of the circular trajectory in the presence of a moving target and time-varying background wind.

## 7. Simulation Results

Firstly, in Section 7.1, scenario 1, that contains two UAVs and a single target, is used to verify the feasibility of the CSTTA in the presence of a moving target and background wind.

Secondly, to verify the performance of the CVEA, the simulation results of the CVEA approach are compared with those using the method used in Ref. [31]. In Section 7.2, comparative experiments are carried out in scenario 2, containing a single UAV and a target.

Finally, Section 7.3 presents scenario 3, that contains three UAVs and a single target. The performance of the proposed TPSA and the SPSA method presented in Ref. [25] are compared to verify that the TPSA has smaller separation errors and a faster convergence rate.

### 7.1. Scenario 1: Tracking a Moving Target Using Two UAVs

In scenario 1, there are two UAVs to perform the cooperative standoff tracking mission for a ground-based moving target in unknown background wind. The simulation conditions for scenario 1 are shown in Table 1. The kinematic constraints of the UAV are shown in Table 2. Table 3 list the initial settings of the UAVs.

**Table 1.** Simulation conditions (scenario 1).

| Parameter | Description | Value |
|:---:|:---:|:---:|
| $t_0$ | Initial time of simulation | 0 s |
| $\Delta t$ | Sampling time | 1.0 s |
| $t_f$ | Final time of simulation | 400 s |
| $T^*$ | Upper bound of the composition velocity | 25 m/s |
| $v_{sd}$ | Desired standoff airspeed | 100 m/s |
| $r_d$ | Desired standoff radius | 1500 m |

**Table 2.** Kinematic constraints of the UAV (scenario 1).

| Parameter | Description | Value |
|:---:|:---:|:---:|
| $v_{min}$ | Allowable minimum airspeed | 60 m/s |
| $v_{max}$ | Allowable maximum airspeed | 160 m/s |
| $\Delta v$ | Unit increment of airspeed | 30 m/s |
| $\omega_{max}$ | Maximum heading rate | 30°/s |

**Table 3.** The initial settings of UAVs (scenario 1).

| UAV $A_i$ | Position $(x_i, y_i)$/m | Velocity $v_i$/(m/s) | Heading $\psi_i$/(°) |
|---|---|---|---|
| $A_1$ | (700, 400) | 100 | 135 |
| $A_2$ | (−1200, 600) | 100 | 20 |

To verify the performance of the CSTTA method in different simulation conditions with respect to the target's motion and background wind, two groups of experiments are carried out.

- Group A: Constant velocities of target and wind;
- Group B: Time-varying velocities of target and wind.

7.1.1. Group A: Constant Velocities of Target and Wind

In group A, a constant velocity (CV) model is used to describe the motion of the ground-based target with constant velocity. The discretized equation for the CV model is expressed as

$$x_t(k+1) = F(k)x_t(k) + G(k)\omega(k)$$

$$F(k) = \begin{bmatrix} 1 & T_s & 0 & 0 \\ 0 & 1 & 0 & 0 \\ 0 & 0 & 1 & T_s \\ 0 & 0 & 0 & 1 \end{bmatrix}; G(k) = \begin{bmatrix} \frac{T_s^2}{2} & T_s & 0 & 0 \\ 0 & 0 & \frac{T_s^2}{2} & T_s \end{bmatrix}^T \tag{71}$$

where $x_t(k) = (x_t, \dot{x}_t, y_t, \dot{y}_t)^T$. $F(k)$ is the state transition matrix and $G(k)$ is the process noise input matrix. The covariance matrix of the process noise $\omega(k)$ is $Q(k) = diag(\delta_x^2(k), \delta_y^2(k))$. $\delta_x(k)$ and $\delta_y(k)$ are the standard deviations related to the target's velocity towards the x- and y-axes. The initial settings of the target are listed in Table 4. The wind velocity is $(w_x, w_y) = (-5, -2)$ m/s.

**Table 4.** The initial settings of target (scenario 1: group A).

| Parameter | Description | Value |
|---|---|---|
| $(x_t, y_t)$ | Initial position of target | (0, 0) m |
| $(\dot{x}_t, \dot{y}_t)$ | Initial velocity of target | (2, 3) m/s |
| $(\delta_x, \delta_y)$ | The standard deviations of target velocity | (0.1, 0.1) m/s |

If the wind velocity is $(w_x, w_y) = (-5, -2)$ m/s, and the target velocity is $(\dot{x}_t, \dot{y}_t) = (2, 3)$ m/s, then the composition velocity is $(T_x, T_y) = (7, 5)$ m/s. On one hand, according to Theorem 2 in Section 3.2, if $v_{sd} = 100$ m/s, $(T_x, T_y) = (7, 5)$ m/s, and $\omega_{max} = 0.524$ rad/s, the minimum allowable standoff radius is $r_{d,min} \approx 901$ $m$. The desired standoff distance $r_d = 1500$ m $> r_{d,min}$. On the other hand, when $v_{sd} = 100$ m/s, $v_{min} = 60$ m/s, $v_{max} = 160$ m/s, and $\Delta v = 30$ m/s, it is verified that the equations $v_{sd} - \Delta v > \sqrt{T_x^2 + T_y^2}$ and $v_{min} + \Delta v \leq v_{sd} \leq v_{max} - \Delta v$ are satisfied. According to Theorem 4 in Section 4.1, the simulation conditions in group A satisfy the input constraints of the UAV.

The trajectories of the UAVs and the target in the inertial frame are shown in Figure 7. The symbol "$T$" represents the target, the black solid line represents the target's trajectory, and the arrow on the trajectory represents the target's motion direction. The red dotted line represents the trajectory of $A_1$, and the blue dash-dotted line represents the trajectory of $A_2$. The triangular arrow "$\triangle$" indicates the heading $\psi$ of the UAV. $A_1(t_0)$ and $A_1(t_f)$ denote the initial and final positions of $A_1$, and $A_2(t_0)$ and $A_2(t_f)$ denote the initial and final positions of $A_2$, respectively. The black arrow "$\rightarrow$" indicates the direction of the wind velocity.

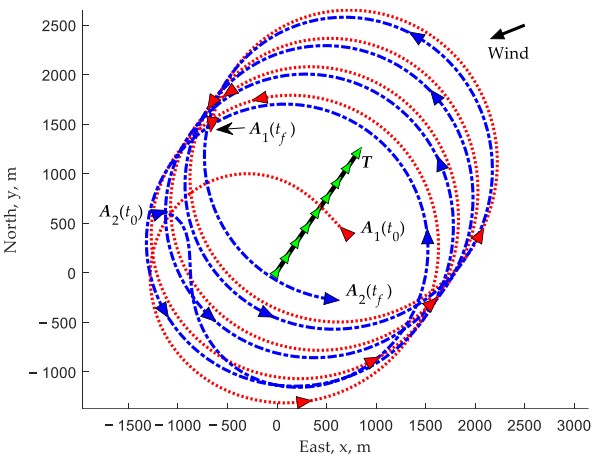

**Figure 7.** Trajectories of UAVs and target in inertial coordinates (scenario 1: group A).

The trajectories of the UAVs in the local target frame coordinates are shown in Figure 8, and the triangular arrow "△" represents the relative course $\chi$ of the UAVs. The UAVs initially have an arbitrary position and heading, and are eventually steered to hover around the target with a predefined standoff distance $r_d$ = 1500 m. Meanwhile, the UAVs are converged to the desired relative course along the LGVF. In addition, the UAVs achieve the desired phase separation when all vehicles arrive at the standoff circle.

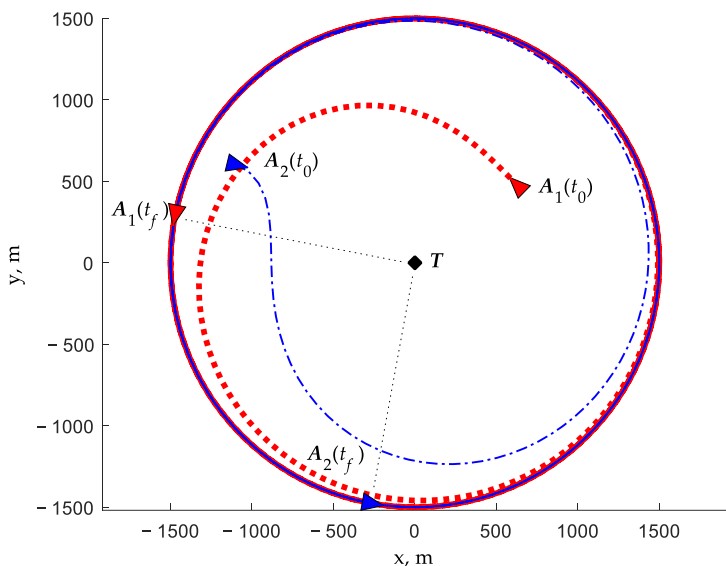

**Figure 8.** Trajectories of UAVs in local target frame coordinates (scenario 1: group A).

The heading rate control input $u$ and airspeed control input $v_s$ are shown in Figures 9 and 10, respectively. It can be seen that $|u_i| \leq \omega_{max}$ and $v_{min} \leq v_{s,i} \leq v_{max}$ ($i = 1, 2$). This implies that the proposed heading rate controller and airspeed controller both satisfy the input constraints of the UAV. The simulation results confirm the conclusions of Theorem 2 and Theorem 4.

The estimated results of the composition velocity, $\left(\hat{T}_x, \hat{T}_y\right)$, are shown in Figure 11. And the corresponding estimated errors $\left(\tilde{T}_x, \tilde{T}_y\right)$ are presented in Figure 12. In order to show more clearly the curves of $\left(\hat{T}_x, \hat{T}_y\right)$ and $\left(\tilde{T}_x, \tilde{T}_y\right)$, Figures 11 and 12 only plot the histories of $\left(\hat{T}_x, \hat{T}_y\right)$ and $\left(\hat{T}_x, \hat{T}_y\right)$ in the time period 0~16 s. It can be seen that the estimated errors converge to 0 at about 8 s, which implies that the proposed estimator can obtain stable and accurate estimates of the composition velocity.

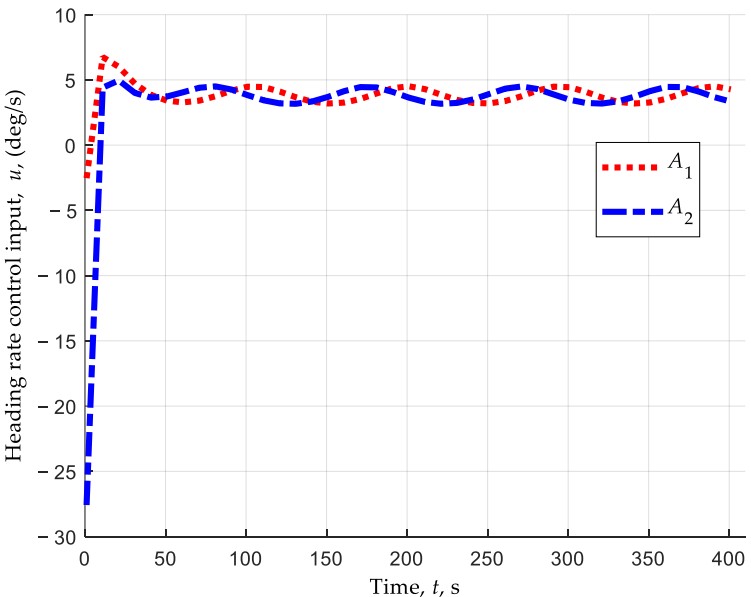

**Figure 9.** Heading rate control inputs of UAVs, *u* (scenario 1: group A).

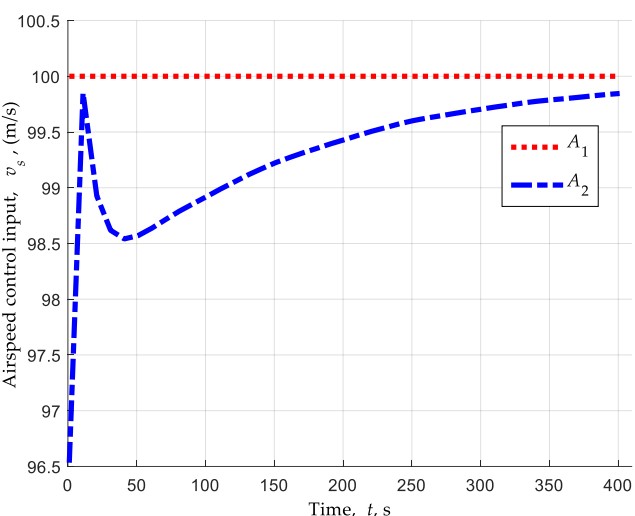

**Figure 10.** Airspeed control inputs of UAVs, $v_s$ (scenario 1: group A).

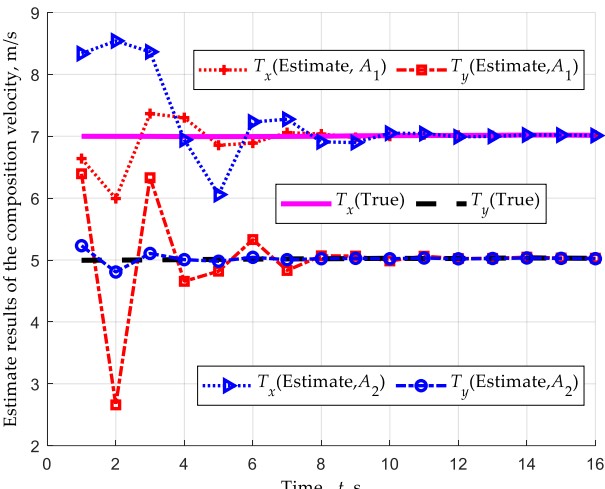

**Figure 11.** Estimated results of the composition velocity, $(\hat{T}_x, \hat{T}_y)$ (scenario 1: group A).

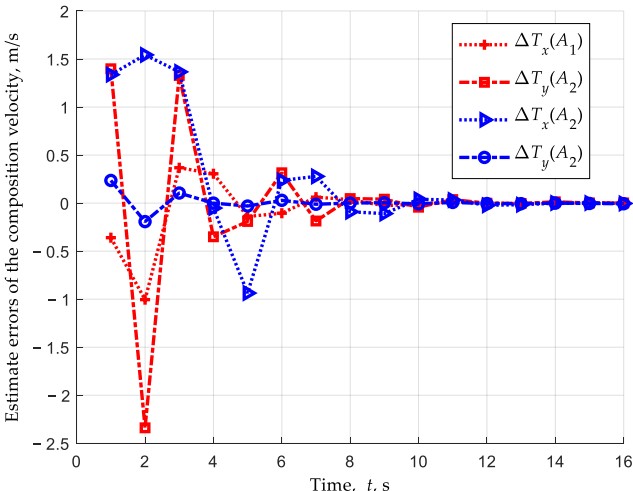

**Figure 12.** Estimated errors of the composition velocity, $\left(\tilde{T}_x, \tilde{T}_y\right)$ (scenario 1: group A).

The relative distances $r_i (i = 1, 2)$ between $A_1$ and the target and $A_2$ and the target are shown in Figure 13. It can be seen that the relative distances of $A_1$ and $A_2$ converge to the desired standoff radius $r_d = 1500$ m. The phase separation angle $\Delta\theta = \theta_2 - \theta_1$ is shown in Figure 14. It can be seen that the phase separation angle $\Delta\theta$ eventually converges to $90°$. This implies that the optimal observation configuration of the UAVs is generated and maintained.

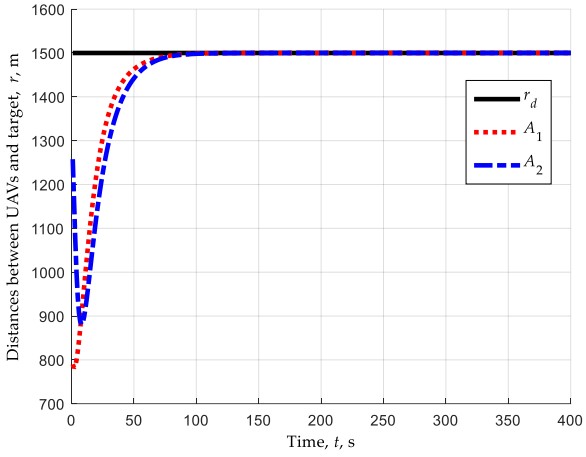

**Figure 13.** Distances between UAVs and target, $r$ (scenario 1: group A).

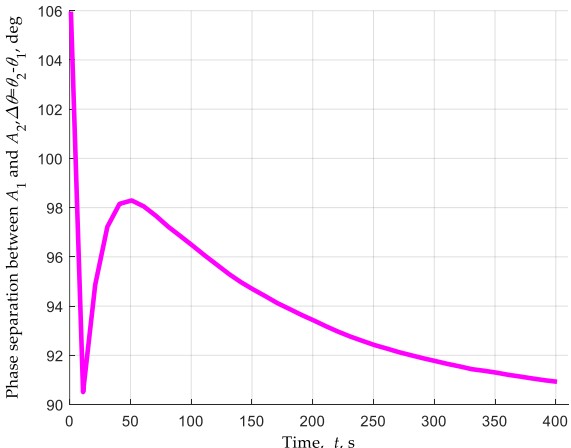

**Figure 14.** Phase separation between $A_1$ and $A_2$, $\Delta\theta = \theta_2 - \theta_1$ (scenario 1: group A).

7.1.2. Group B: Time-Varying Velocities of Target and Wind

A ground-based moving target has a low velocity, and they often and irregularly perform stop-and-go maneuvers with a much smaller turn radius. More agile target maneuvers are likely to have more significant higher-order derivatives, such as the third derivative of the target position, i.e., the acceleration rate or jerk of the target. A lower-order tracking model, such as the CV model, cannot adequately handle higher-order derivatives. Hence, a good model to apply to tracking a ground-based target is the jerk model. The jerk model defines the target acceleration as a correlated process exponentially decreasing in time. A discretized system equation for the jerk model for a ground target is thus expressed in the following form [16]:

$$x_t(k+1) = F(k)x_t(k) + \eta(k) \tag{72}$$

where $x_t(k) = \left(x_t, \dot{x}_t, \ddot{x}_t, y_t, \dot{y}_t, \ddot{y}_t\right)^T$. The state transition matrix $F(k)$ can be represented as follows:

$$F(k) = \begin{bmatrix} 1 & T_s & \left(e^{-\alpha T_s} + \alpha T_s - 1\right)/\alpha^2 & 0 & 0 & 0 \\ 0 & 1 & \left(1 - e^{-\alpha T_s}\right)/\alpha & 0 & 0 & 0 \\ 0 & 0 & e^{-\alpha T_s} & 0 & 0 & 0 \\ 0 & 0 & 0 & 1 & T_s & \left(e^{-\alpha T_s} + \alpha T_s - 1\right)/\alpha^2 \\ 0 & 0 & 0 & 0 & 1 & \left(1 - e^{-\alpha T_s}\right)/\alpha \\ 0 & 0 & 0 & 0 & 0 & e^{-\alpha T_s} \end{bmatrix} \tag{73}$$

where $\alpha$ is a correlation parameter that allows for the modeling of different classes of targets: small $\alpha$ for targets with relatively slow maneuvers, and large $\alpha$ for targets with fast and evasive maneuvers. The covariance matrix of the process noise $\eta(k)$ can be modeled as follows:

$$Q(k) = \frac{\sigma_a^2}{\alpha^4} \begin{bmatrix} q_{11} & q_{12} & q_{13} & 0 & 0 & 0 \\ q_{12} & q_{22} & q_{23} & 0 & 0 & 0 \\ q_{13} & q_{23} & q_{33} & 0 & 0 & 0 \\ 0 & 0 & 0 & q_{11} & q_{12} & q_{13} \\ 0 & 0 & 0 & q_{12} & q_{22} & q_{23} \\ 0 & 0 & 0 & q_{13} & q_{23} & q_{33} \end{bmatrix} \tag{74}$$

where $\sigma_a$ is the standard deviation related to the target's acceleration, and the definitions of $q_{ij}$ are

$$\begin{aligned} q_{11} &= \left(1 - l + 2m + \tfrac{2}{3}m^3 - 2m^2 - 4m\sqrt{l}\right) \\ q_{12} &= \alpha\left(l + 1 - \tfrac{1}{l} - 2m + m^2\right) \\ q_{13} &= \alpha^2\left(1 - l - 2m\sqrt{l}\right) \\ q_{22} &= \alpha^2\left(4\sqrt{l} - 3 - l + 2m\right) \\ q_{23} &= \alpha^3\left(l + 1 - 2\sqrt{l}\right) \\ q_{33} &= \alpha^4(1 - l) \end{aligned} \tag{75}$$

where $l = e^{-2\alpha T_s}$ and $m = \alpha T_s$. The initial settings of the target are listed in Table 5.

**Table 5.** The initial settings of target (scenario 1: group B).

| Parameter | Description | Value |
|---|---|---|
| $(x_t, y_t)$ | Initial position of target | $(0, 0)$ m |
| $(\dot{x}_t, \dot{y}_t)$ | Initial velocity of target | $(2, 3)$ m/s |
| $\alpha$ | Correlation parameter | 0.6 |
| $\sigma_a$ | The standard deviations of target acceleration | 0.66 m/s$^2$ |

The characteristics of the trajectory and moving behavior of the target which are described by the jerk model are illustrated here through a numerical simulation. Figure 15 shows the trajectory of the moving target based on the jerk model. The time-varying speed and direction of the moving target based on the jerk model are shown in Figures 16 and 17, respectively. It is observed that the jerk is not negligible, and thus the target is maneuvering with time.

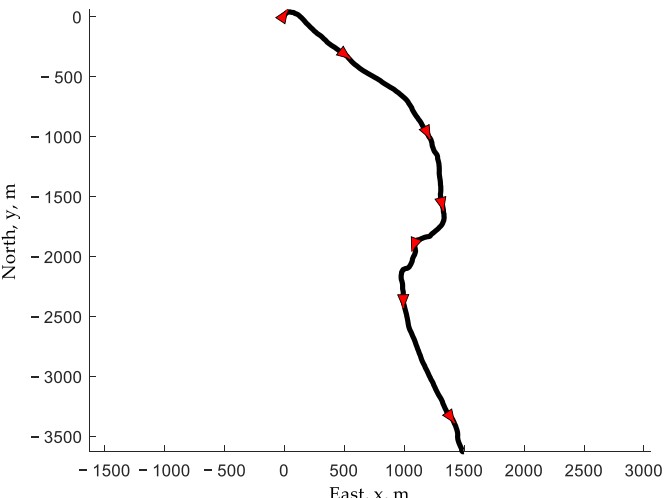

**Figure 15.** Trajectory of moving target with jerk model (group B).

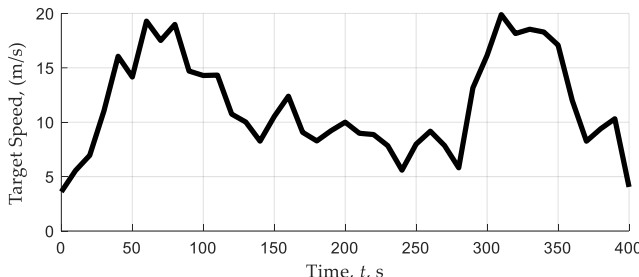

**Figure 16.** Speed of moving target with jerk model (scenario 1: group B).

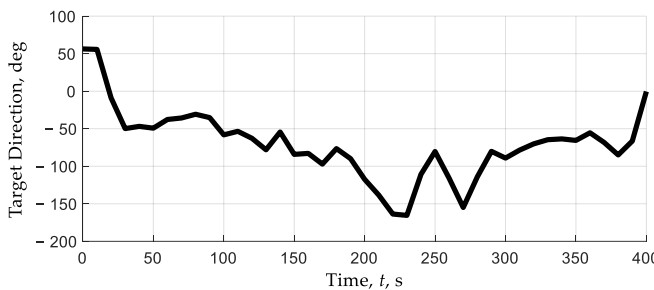

**Figure 17.** Direction of moving target with jerk model (scenario 1: group B).

It is worth noting that according to Theorem 2 and Theorem 4, the upper bound of the target speed must be restricted. In other words, in order to meet the input constraints of the UAV, the maximum speed of the target is not more than 20 m/s.

In addition, if the wind is time-varying, let us consider an example of a variable wind case, where the wind model is similar to the model used in [26], which is given below:

$$
\begin{cases}
W_x = v_w cos[\omega(t - t_0) + \phi_0] \\
W_y = v_w sin[\omega(t - t_0) + \phi_0]
\end{cases}
\tag{76}
$$

where the constants are $v_w = 5$ m/s, $\omega = (\pi/180)$ rad/s, $t_0 = 0$ s, and $\phi_0 = 30°$.

From the settings of the target and the wind, it can be seen that, since the target speed is not more than 20 m/s and the wind speed is not more than 5 m/s, the upper bound of the composition velocity is $T^* = 25$ m/s in Table 1. According to Theorem 2, if $v_{sd} = 100$ m/s, $T^* = 25$ m/s, and $\omega_{max} = 0.524$ rad/s, the minimum allowable standoff radius is $r_{d,min} \approx 1193.8$ m. The desired standoff distance, $r_d = 1500$ m $> r_{d,min}$, satisfies the heading rate constraint. In addition, it is verified that the inequalities $v_{sd} - \Delta v > \sqrt{T_x^2 + T_y^2}$ and $v_{min} + \Delta v \leq v_{sd} \leq v_{max} - \Delta v$ are satisfied. According to Theorem 4, the simulation conditions in group B satisfy the input constraints of the UAV.

The trajectories of the UAVs and the target in the inertial frame are shown in Figure 18. The trajectories of the UAVs in the local target frame coordinates are shown in Figure 19. The UAVs are eventually steered to hover around the target at the predefined standoff distance $r_d = 1500$ m, and achieve the desired phase separation on the standoff circle.

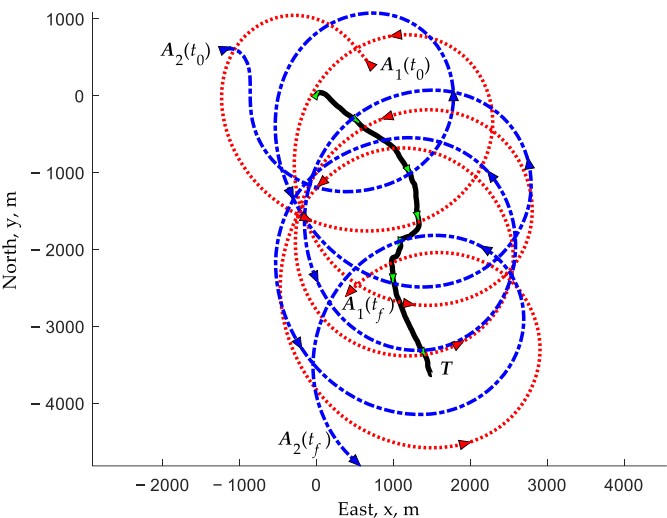

**Figure 18.** Trajectories of UAVs and target in inertial coordinates (scenario 1: group B).

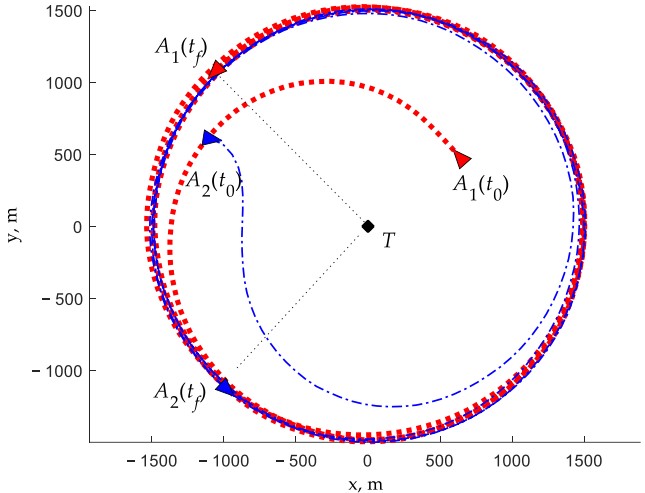

**Figure 19.** Trajectories of UAVs in local target frame coordinates (scenario 1: group B).

The heading rate control input $u$ and airspeed control input $v_s$ are shown in Figures 20 and 21, respectively. The control inputs satisfy the kinematic constraints of the UAVs.

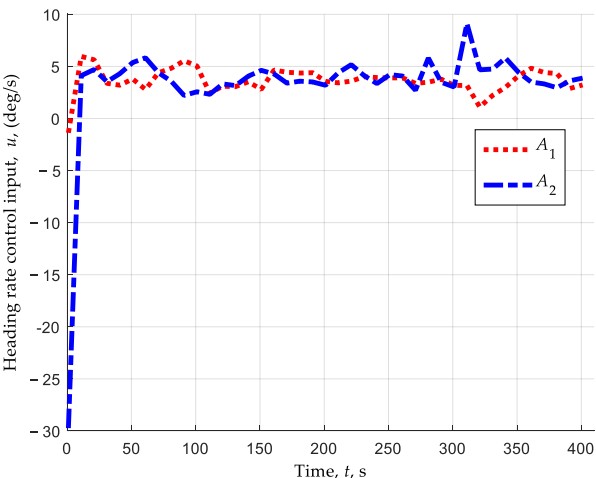

**Figure 20.** Heading rate control inputs of UAVs, *u* (scenario 1: group B).

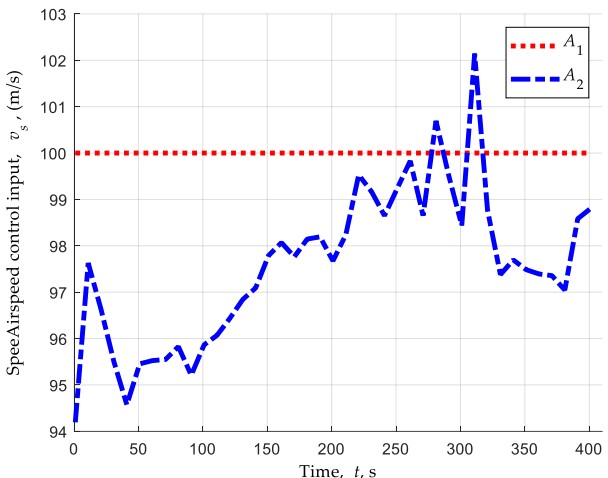

**Figure 21.** Airspeed control inputs of UAVs, $v_s$ (scenario 1: group B).

The estimated results of the composition velocity, $(\hat{T}_x, \hat{T}_y)$, are shown in Figures 22 and 23, and the corresponding estimated errors $(\tilde{T}_x, \tilde{T}_y)$ are presented in Figures 24 and 25, respectively. It can be seen that the proposed CVEA still obtains stable and accurate estimates of the composition velocity in the presence of time-varying velocities of a ground-based moving target and background wind.

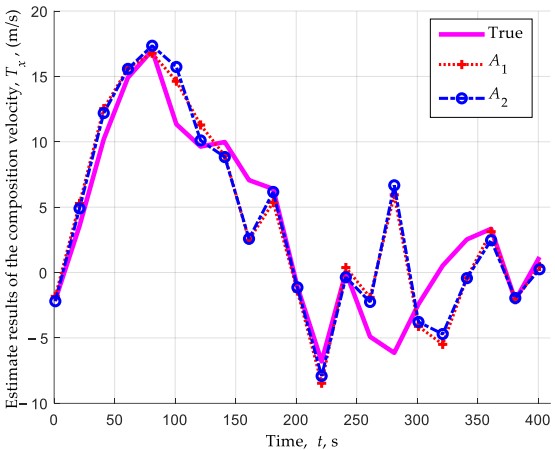

**Figure 22.** Composition velocity estimate results, $\hat{T}_x$ (scenario 1: group B).

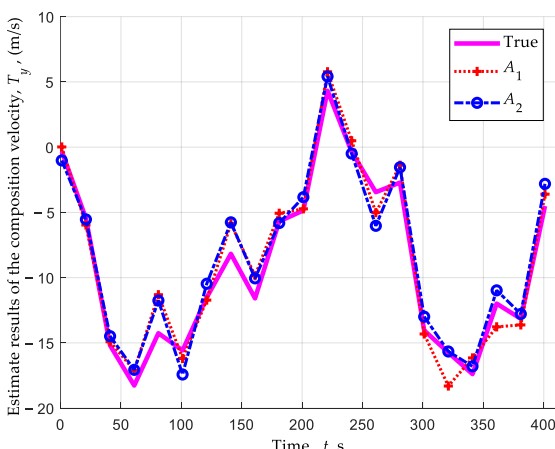

**Figure 23.** Composition velocity estimate results, $\hat{T}_y$ (scenario 1: group B).

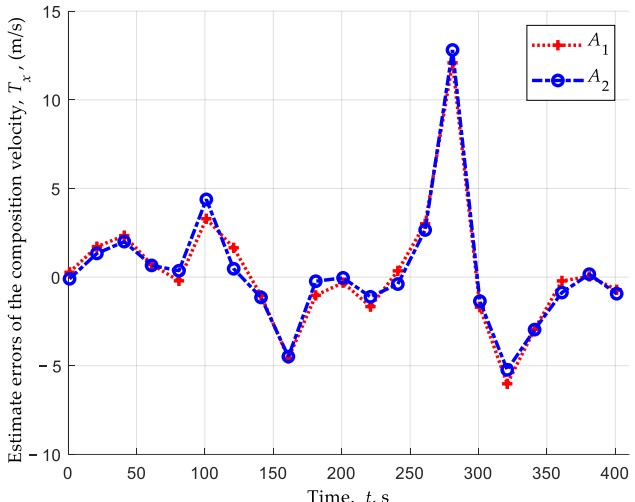

**Figure 24.** Composition velocity estimate errors, $\tilde{T}_x$ (scenario 1: group B).

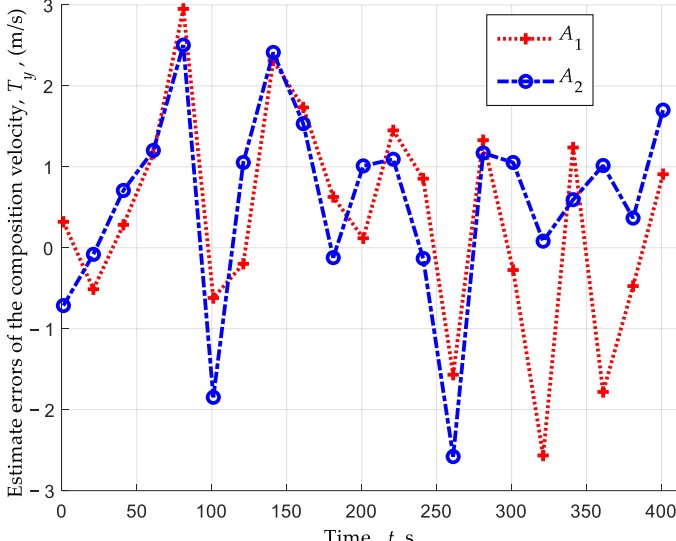

**Figure 25.** Composition velocity estimate errors, $\tilde{T}_y$ (scenario 1: group B).

The relative distances $r_i(i = 1, 2)$ between $A_1$ and the target and $A_2$ and the target are shown in Figure 26. It can be seen that there exist errors between the real relative distances

and the desired value when the target performs greater maneuvers, for example, in the time period 300~350 s. However, the UAVs can still converge to the neighborhood of the desired orbit with small distance errors. The phase separation angle $\Delta\theta = \theta_2 - \theta_1$ is shown in Figure 27. Due to the target's maneuvers, $A_2$ tries to adjust its airspeed to form the desired phase separation with $A_1$, and thus the phase separation angle $\Delta\theta$ oscillates and converges to the neighborhood of the desired value.

Therefore, the simulation results in group B verify the performance of the proposed CSTTA approach. The CSTTA approach can guarantee that the UAVs perform successfully coordinated standoff tracking in the presence of time-varying velocities of a ground-based moving target and background wind.

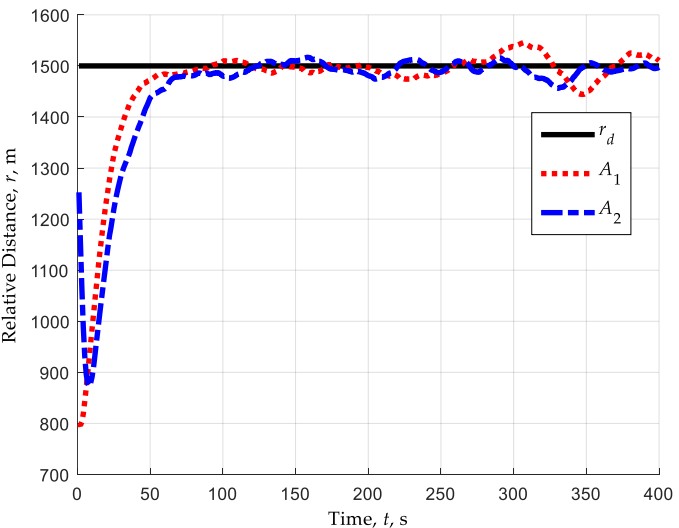

**Figure 26.** Distances between UAVs and target, $r$ (scenario 1: group B).

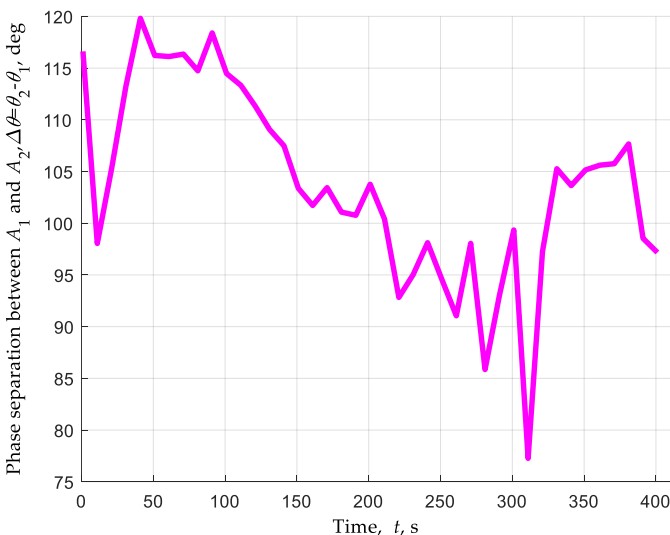

**Figure 27.** Phase separation between $A_1$ and $A_2$, $\Delta\theta = \theta_2 - \theta_1$ (scenario 1: group B).

### 7.2. Scenario 2: Tracking a Moving Target Using a Single UAV

In scenario 2, a single UAV $A_1$ is used to execute the standoff target tracking mission. The initial settings of $A_1$ are listed in Table 6, and the other simulation conditions are the same as for group B in scenario 1. In order to verify that the proposed CVEA can generate an effective and accurate composition velocity estimation to effectively improve the standoff tracking performance, the simulation results of the single drone's standoff

target tracking algorithm (STTA), as shown in Equation (24) with the CVEA (denoted as "STTA+CVEA"), are compared with those of the method reported in [31].

**Table 6.** Initial settings of UAVs (scenario 2).

| UAV $A_i$ | Position $(x_i,y_i)$/m | Velocity $v_i$/(m/s) | Heading $\psi_i$/(°) |
|---|---|---|---|
| $A_1$ | (600, 200) | 100 | 90 |

Firstly, we need to define several performance indexes to evaluate the effectiveness of the CVEA approach and the method reported in [31]. Thus, the global average errors (GAEs) are introduced. It is worth noting that global means it is averaged over the *N* UAVs. This definition is not only appropriate for the scenario of a single UAV, but also for the scenario of multiple UAVs. At time $t_k$, the GAE of the relative distance regulation is defined as $e_r(t_k) = \frac{1}{N}\sum_{i=1}^{N}|r_i(t_k) - r_d|$. Similarly, the GAE of the relative course tracking is defined as $e_\chi(t_k) = \frac{1}{N}\sum_{i=1}^{N}\left|\chi_i(t_k) - \chi_i^d(t_k)\right|$; the GAE of the phase separation is defined as $e_\theta(t_k) = \frac{1}{N}\sum_{i=1}^{N}|\theta_j(t_k) - \theta_i(t_k) - \theta_d|$; the GAE of the composition velocity estimation is defined as $e_T(t_k) = \frac{1}{N}\sum_{i=1}^{N}\left(\tilde{T}_x^2 + \tilde{T}_y^2\right)^{\frac{1}{2}}$. These GAEs can be used to analyze the convergence of the designed controllers and estimator, including the heading rate controller proposed in Section 3.2, the airspeed controller proposed in Section 4.1, and the composition velocity estimator proposed in Section 5.1.

Then, in order to analyze the convergence rate of the designed controllers and estimator, the integrated time absolute error (ITAE) is defined as follows:

$$J_{ITAE}(e) = \int_0^t \tau \cdot e(\tau)d\tau \tag{77}$$

where $e(\tau)$ represents the GAEs at time $\tau$, which includes the GAE of relative distance regulation, the GAE of relative course tracking, the GAE of phase separation, and the GAE of composition velocity estimation. It can be observed from Equation (77) that a smaller $J_{ITAE}(e)$ implies a smaller convergence error $e(\tau)$ and faster convergence rate.

Due to the process noise in the target's motion, the results of a single simulation are not sufficient to illustrate that the proposed method is effective. Hence, we run a Monte Carlo simulation 300 times for scenario 2 to further analyze the performances of the two methods using the above-defined GAEs and ITAEs.

Figure 28 shows the performance of the relative distance regulation. Figure 29 shows the performance of the relative course tracking. It can be seen that the proposed "STTA+CVEA" method has smaller GAEs and ITAEs in relative distance regulation and relative course tracking than that in [31]. This implies that the "STTA+CVEA" method has better performance than the method presented in [31] in the standoff target tracking mission of a single UAV. The reason for the above phenomenon is that in contrast to [31], in which only the radial distance of the offset is used, the "STTA+CVEA" method updates the estimation results according to the offset vector, containing more feedback information, and thus our proposed method produces a better composition velocity estimation to enhance the tracking performance of the UAV.

Figure 30 shows the performance of the composition velocity estimation. The GAE and ITAE in the composition velocity estimation produced by "STTA+CVEA" are both smaller than that in the method presented in [31]. In other words, the estimates generated in our approach can converge to the true values more quickly with smaller convergence errors.

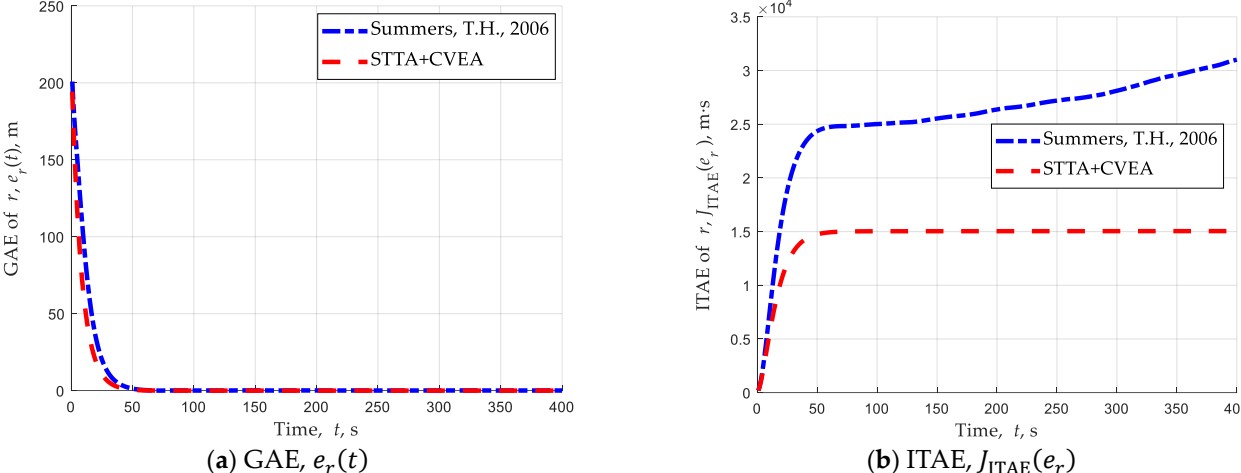

**Figure 28.** Performance of relative distance regulation (scenario 2). (**a**) Global average error in relative distance. (**b**) Integrated time absolute error in relative distance [31].

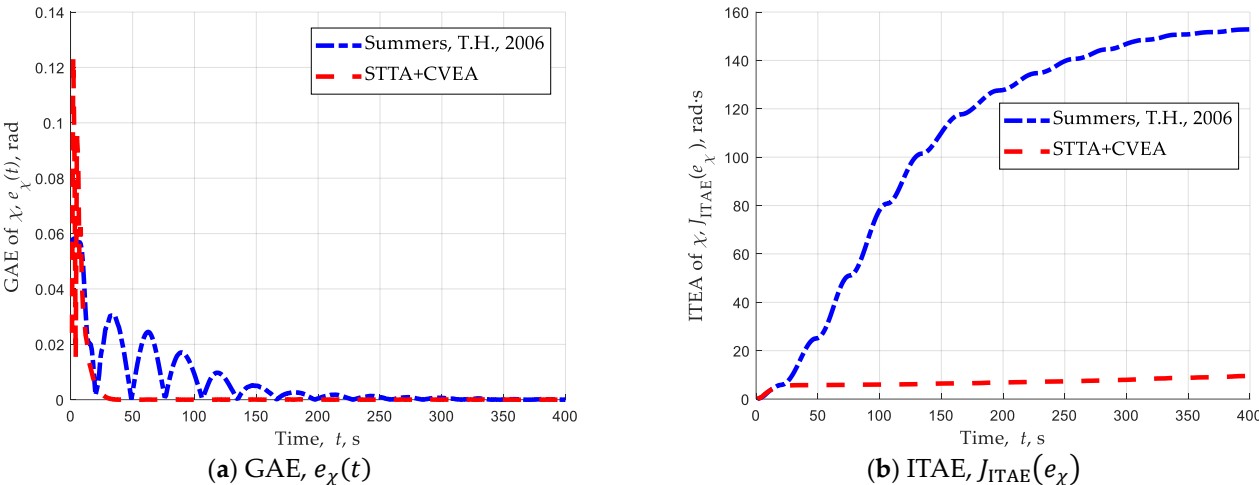

**Figure 29.** Performance of relative course tracking (scenario 2). (**a**) Global average error in relative course. (**b**) Integrated time absolute error in relative course [31].

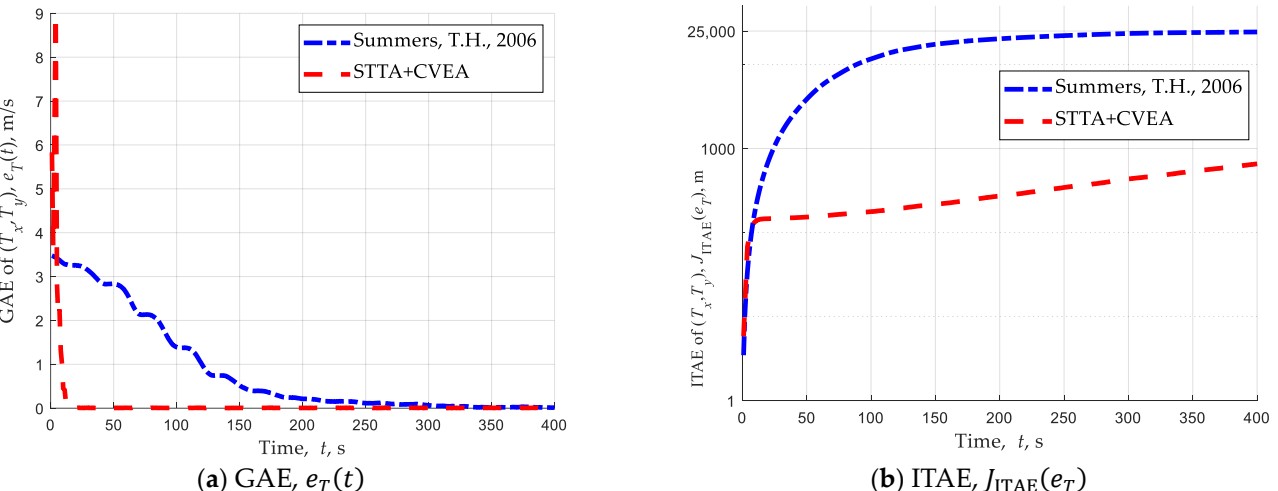

**Figure 30.** Performance of composition velocity estimation (scenario 2). (**a**) Global average error in composition velocity estimation. (**b**) Integrated time absolute error in composition velocity estimation [31].

Figure 31 shows a comparison of the heading rate control inputs generated using the two methods. Figure 32 shows a comparison of the relative course errors generated using the two methods. In order to show more clearly the curves of $u$ and $\chi_e$, Figures 31 and 32 only plot the histories of $u$ and $\chi_e$ in the time period 0~150 s. It can be seen that the CVEA can obtain the composition velocity estimate results accurately and quickly compared to the results in [31]; the heading rate control input signal is smooth and the relative course error converges to 0 without oscillation. This confirms that the CVEA can improve the effectiveness and the robustness of the standoff target tracking algorithm (STTA) for the UAV in unknown background wind.

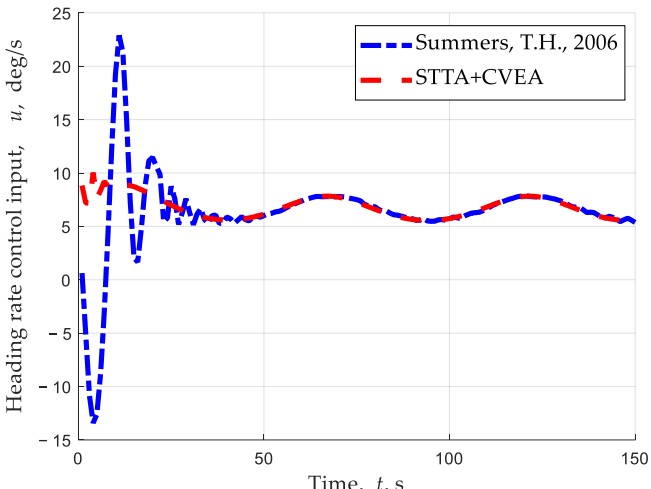

**Figure 31.** Heading rate control input, $u$ (scenario 2) [31].

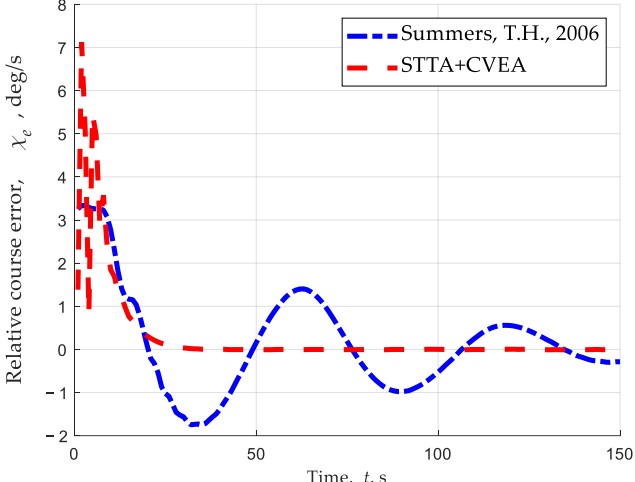

**Figure 32.** Relative course error, $\chi_e$ (scenario 2 [31]).

A performance comparison of "STTA+CVEA" and Ref. [31] is shown in Table 7. Compared to [31], the GAE and ITAE of the relative distance regulation are reduced by 29.44% and 43.84%, respectively, in the proposed "STTA+CVEA" method. The GAE and ITAE in the relative course tracking are reduced by 67.74% and 93.5%, respectively. The GAE and ITAE in the composition velocity estimation are reduced by 88.90% and 98.14%, respectively. Therefore, in the standoff target tracking mission using a single UAV in unknown background wind, our method has better performance than the method presented in [31].

**Table 7.** Performance comparison of "STTA+CVEA" and Ref. [31] (scenario 2).

| Control Objective | Performance Index | STTA+CVEA | Ref. [31] | Percentage/(%) |
|---|---|---|---|---|
| Relative distance regulation | $e_r(t)/m$ | 4.2229 | 5.9846 | 29.44 |
| | $J_{ITAE}(e_r)/(m \times s)$ | $1.4436 \times 10^4$ | $2.5706 \times 10^4$ | 43.84 |
| Relative course tracking | $e_\chi(t)/rad$ | 0.0020 | 0.0062 | 67.74 |
| | $J_{ITAE}(e_\chi)/(rad \times s)$ | 6.9507 | 106.9023 | 93.50 |
| Composition velocity estimation | $e_T(t)/(m \times s^{-1})$ | 0.0954 | 0.8598 | 88.90 |
| | $J_{ITAE}(e_T)/(m)$ | 313.6534 | $1.6874 \times 10^4$ | 98.14 |

*7.3. Scenario 3: Tracking a Moving Target Using Three UAVs*

In scenario 3, three UAVs are used to execute a standoff target tracking mission. Table 8 lists the initial settings of $A_1$, $A_2$, and $A_3$, and the other simulation conditions are the same as for group B in scenario 1. In order to verify that the proposed TPSA approach has smaller separation errors and a faster convergence rate, and thus improves the cooperative tracking performance of the UAVs, the simulation results of the TPSA are compared with the SPSA method presented in [25].

**Table 8.** Initial settings of UAVs (scenario 3).

| UAV $A_i$ | Position $(x_i, y_i)$/m | Velocity $v_i$/(m/s) | Heading $\psi_i$/(°) |
|---|---|---|---|
| $A_1$ | (600, 200) | 100 | 90 |
| $A_2$ | (−600, 300) | 100 | 120 |
| $A_3$ | (0, 800) | 100 | −170 |

It is important to note that in [25] they only study the case of two collaborative UAVs tracking a ground target with background wind. Fortunately, the method in [25] is also suitable for the team of $N$ UAVs; more details are described in [22].

In addition, a robust term is introduced to obtain disturbance rejection for wind gusts in [25]. The GAE ($e_T(t_k)$) and the corresponding ITAE ($J_{ITAE}(e_T)$) are not suitable to evaluate the effectiveness of the method used in [25]. Therefore, we mainly focus on the phase separation problem solved by implementing the proposed TPSA and the SPSA in [25].

Figure 33 shows the performance of intervehicle phase separation in the process of the cooperative standoff target tracking mission using three UAVs. Due to the discontinuity of the wrapped space separation angle leading to oscillations in the control input signal, the curve of $e_\theta(t)$ in the SPSA method shown in Figure 33a fluctuates in the convergence process. The fluctuation results in a slow convergence rate, which is confirmed by the curve of $J_{ITAE}(e_\theta)$ in the SPSA method shown in Figure 33b. In contrast to the SPSA method, the curve of $e_\theta(t)$ in our proposed TPSA approach smoothly converges to 0. In summary, comparing to the SPSA method described in [25], our proposed TPSA has a smaller phase separation error and faster convergence rate.

The comparison results for other performance indexes are shown in Table 9. Comparing to [25], the GAE and ITAE of the intervehicle phase separation are reduced by 22.51% and 4.85%, respectively, in our proposed TPSA method. From Table 9, it is verified that in the cooperative standoff target tracking mission using multiple UAVs in unknown background wind, our proposed CSTTA method has better performance than the method presented in [25].

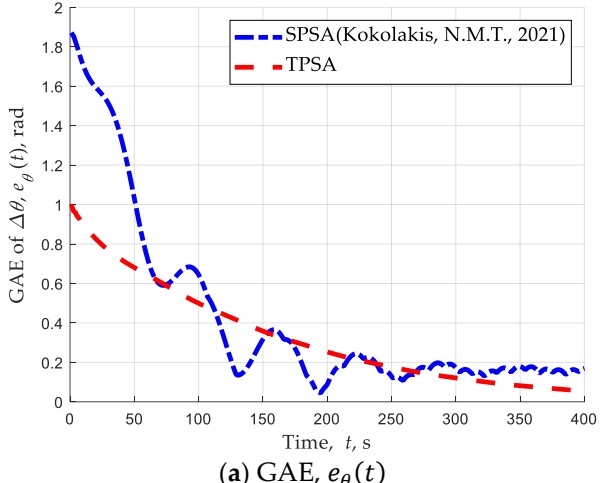
(**a**) GAE, $e_\theta(t)$

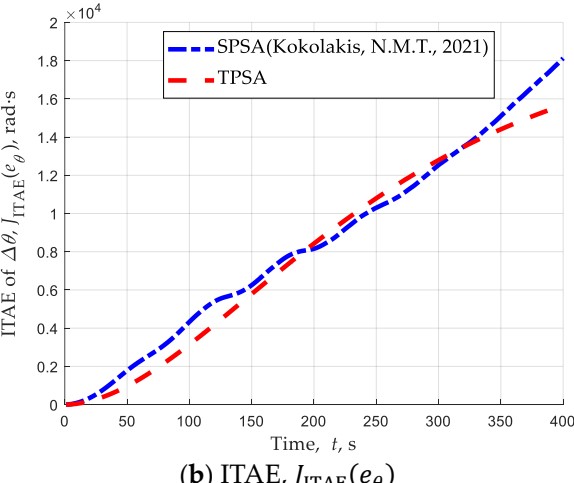
(**b**) ITAE, $J_{\text{ITAE}}(e_\theta)$

**Figure 33.** Performance of intervehicle phase separation (scenario 3). (**a**) Global average error in phase separation. (**b**) Integrated time absolute error in phase separation [25].

**Table 9.** Performance comparison of CSTTA and Ref. [25] (scenario 3).

| Control Objective | Performance Index | CSTTA | Ref. [25] | Percentage/(%) |
|---|---|---|---|---|
| Relative distance regulation | $e_r(t)/m$ | 6.5996 | 10.7001 | 38.32 |
| | $J_{ITAE}(e_r)/(m \times s)$ | $2.5579 \times 10^4$ | $5.7246 \times 10^4$ | 55.32 |
| Relative course tracking | $e_\chi(t)/rad$ | 0.0090 | 0.0135 | 33.33 |
| | $J_{ITAE}(e_\chi)/(rad \times s)$ | 21.3996 | 117.3008 | 81.76 |
| Intervehicle phase separation | $e_\theta(t)/(rad)$ | 0.3311 | 0.4273 | 22.51 |
| | $J_{ITAE}(e_\theta)/(rad \times s)$ | $8.0284 \times 10^3$ | $8.4378 \times 10^3$ | 4.85 |

## 8. Conclusions

This paper investigates the standoff tracking of a ground-based moving target using multiple fixed-wing UAVs in unknown background wind. The main contribution of this paper is to develop a cooperative standoff target tracking algorithm (CSTTA), which considers the control input constraints of the fixed-wing UAV, the target's motion, and an unknown wind. The following conclusions can be obtained.

(i)   A fundamental cooperative standoff target tracking problem includes three control objectives: relative distance regulation, relative course convergence, and intervehicle phase separation. In addition, in the case of unknown background wind, it is essential to enhance the wind resistance capacity of the UAV.

(ii)  A heading rate control law based on LGVF is introduced to regulate the position of a UAV on a circle around the target with a constant standoff distance. It is proved that the proposed heading rate controller can achieve standoff target tracking for a single UAV in the condition of an arbitrary initial position and heading. Due to the heading rate input being constrained, the predefined standoff distance between the UAV and the target has an allowable lower bound. The minimum allowable standoff distance is formulated in this paper.

(iii) A new temporal phase separation algorithm (TPSA) is proposed to achieve the desired temporal phase separation in a cooperative standoff target tracking mission. The TPSA approach takes into account the minimum and maximum airspeed constraints, and can avoid the discontinuity of wrapped space phase angles. The results of comparison simulations show that the TPSA has a smaller convergence error and faster convergence rate than the previously reported space phase separation method.

(iv)  The offset between the actual vehicle trajectory and the desired LGVF orbit can be utilized to estimate the composition velocity of the target's motion and background

wind. The results of comparison simulations show that the proposed composition velocity estimation algorithm (CVEA) can effectively estimate the composition velocity, and thus enhance the tracking performance of the UAVs in the presence of wind and a moving target.

There are many potential directions for future consideration. Firstly, the current models and algorithms will be extended to three-dimensional coordinates. Secondly, the target location algorithm will be designed to explore the possibility of using an onboard observation sensor such as a camera to facilitate tracking. Thirdly, more complex mission environments will be considered, and the methods of terrain obstacle and intervehicle collision avoidance will be introduced to enhance aircraft safety. The proposed algorithms will be implemented on real UAVs in future works. Finally, the communication constraints, such as limited communication range and communication delays, will be considered.

**Author Contributions:** Conceptualization, Z.L.; data curation, Z.L. and L.X.; formal analysis, L.X.; funding acquisition, Z.Z.; investigation, Z.L.; methodology, Z.L.; project administration, Z.Z.; resources, L.X.; software, Z.L.; supervision, Z.Z.; validation, L.X.; visualization, Z.L.; writing—original draft, Z.L.; writing—review and editing, L.X. All authors have read and agreed to the published version of the manuscript.

**Funding:** This work received funding from the Scientific Research Program of the Education Department of Hubei Provincial Government, under Grant No. D20212902, and the Program of National undergraduate innovation and entrepreneurship, under Grant No. 202210514008.

**Data Availability Statement:** The data presented in this study are available on request from the corresponding author. The data are not publicly available due to privacy restrictions.

**Conflicts of Interest:** The authors declare no conflict of interest.

## Appendix A. Proof of Lemma 1

**Proof:** Construct the following function

$$F(\psi) = \frac{v_r}{\lambda_u(\psi)} \tag{A1}$$

Substituting the relative speed $v_r$ given by Equation (6) and $\lambda_u(\psi)$ given by Equation (9) into Equation (A1) yields

$$F(\psi) = \frac{\left(v_s^2 + T^2 - 2v_s T\xi\right)^{\frac{3}{2}}}{v_s^2 - v_s T\xi} \tag{A2}$$

where $T = \sqrt{T_x^2 + T_y^2}$, $\eta = arctan\left(\frac{T_x}{T_y}\right) \in [-\pi, \pi]$, and $\xi = sin(\psi + \eta) \in [-1, 1]$. Differentiating Equation (A2) with respect to $\xi$, one obtains

$$F_\xi(\psi) = \frac{v_s T\left(v_s^2 + T^2 - 2v_s T\xi\right)^{\frac{1}{2}}\left(T^2 + v_s T\xi - 2v_s^2\right)}{\left(v_s^2 - v_s T\xi\right)^2} \tag{A3}$$

when $\xi \in [-1, 1]$, then $F_\xi(\psi) < 0$. This means that $F(\psi)$ decreases monotonically with increasing $\xi$. Therefore, when $\xi = -1$, $F(\psi)$ reaches a maximum, i.e., $F(\psi) \leq \frac{(v_s+T)^2}{v_s}$. Thus, the proof for Lemma 1 is completed. □

## Appendix B. Proof of Lemma 2

**Proof:** Firstly, two cases are discussed in the proof of Conclusion ①.

(a)  When $\chi_e = 0$, then for all $k_1 > 0$, $\dot{\chi}_d = \frac{4v_r r_d^3}{\left(r^2+r_d^2\right)^2} \geq -\lambda_u(\psi)\omega_{max} + k_1 sin\chi_e$ is always true.

(b)  If $\chi_e \in (0, \pi)$, then we construct the following function

$$F(\chi_e, r) = \frac{1}{sin\chi_e}\left(\dot{\chi}_d + \frac{4v_r}{r_d}\right) \tag{A4}$$

Partially differentiating Equation (A4) with respect to $\chi_e$, one obtains

$$\frac{\partial F(\chi_e, r)}{\partial \chi_e} = \frac{-4v_r}{r_d sin^2\chi_e}\left[\frac{r_d^4}{\left(r^2 + r_d^2\right)^2} + cos\chi_e\right] \tag{A5}$$

It can be observed from Equation (A5) that when $cos\chi_e^* = -\frac{r_d^4}{(r^2+r_d^2)^2} < 0$ it implies $\chi_e^* \in \left(\frac{\pi}{2}, \pi\right)$, then $F(\chi_e, r)$ reaches the maximum $F(\chi_e^*, r)$, which is expressed as follows:

$$F(\chi_e^*, r) = \frac{v_r}{r}\left(\frac{\zeta}{sin\chi_e^*}\right) + 4\frac{v_r}{r_d}\left(\frac{1}{sin\chi_e^*}\right) \tag{A6}$$

where

$$\zeta = sin\phi_{GF}(1 + cos\phi_{GF})cos\chi_e^* + \left(cos\phi_{GF} - sin^2\phi_{GF}\right)sin\chi_e^* \tag{A7}$$

For the first term in Equation (A6), one can deduce

$$\frac{\zeta}{sin\chi_e^*} \geq \frac{r_d^4 - r^4 - 4r^2r_d^2}{\left(r^2 + r_d^2\right)^2} \tag{A8}$$

For the second term in Equation (A6), one can deduce

$$\left(\frac{4v_r}{r_d}\right)\left(\frac{1}{sin\chi_e^*}\right) \geq \left(\frac{4v_r}{r_d}\right)sin\chi_e^* \geq \left(\frac{4v_r}{r_d}\right)\left(\frac{\left[(r^2+r_d^2)^2 - r_d^4\right]^2}{\left(r^2+r_d^2\right)^4}\right)^{\frac{1}{2}}$$
$$= \left(\frac{4v_r}{r_d}\right)\left(\frac{r^4 + 2r^2r_d^2}{\left(r^2+r_d^2\right)^2}\right) \tag{A9}$$

According to Equation (A6), one derives

$$\begin{aligned}F(\chi_e, r) \geq F(\chi_e^*, r) \quad &\geq \frac{v_r}{r}\lambda_2 + \left(\frac{4v_r}{r_d}\right)sin\chi_e^* \\ &\geq \frac{v_r}{r}\left(\frac{r_d^4 - r^4 - 4r^2r_d^2}{\left(r^2+r_d^2\right)^2}\right) + \left(\frac{4v_r}{r_d}\right)\left(\frac{r^4 + 2r^2r_d^2}{\left(r^2+r_d^2\right)^2}\right) \\ &\stackrel{\varepsilon = \frac{r}{r_d}}{\Longrightarrow} \frac{v_r}{r_d}\left(1 + \frac{1 - \varepsilon - 4\varepsilon^2 + 6\varepsilon^3 - \varepsilon^4 + 3\varepsilon^5}{\varepsilon(\varepsilon^2 + 1)^2}\right) > \frac{v_r}{r_d}\end{aligned} \tag{A10}$$

Therefore, let $k_1 = \frac{v_r}{r_d}$, we obtain $F(\chi_e, r) \geq k_1$, and thus

$$\frac{1}{sin\chi_e}\left(\dot{\chi}_d + \frac{4v_r}{r_d}\right) \geq k_1 \Rightarrow \dot{\chi}_d + \frac{4v_r}{r_d} \geq k_1 sin\chi_e \tag{A11}$$

According to the conclusion of Lemma 1 and Equation (26), it can be obtained that

$$\lambda_u(\psi)\omega_{max} \geq \frac{4v_r}{r_d} \tag{A12}$$

Substituting Equation (A12) into Equation (A11) yields

$$\dot{\chi}_d + \lambda_u(\psi)\omega_{max} \geq \dot{\chi}_d + \frac{4v_r}{r_d} \geq k_1 sin\chi_e \Rightarrow \dot{\chi}_d \geq -\lambda_u(\psi)\omega_{max} + k_1 sin\chi_e \tag{A13}$$

The proof of Conclusion ① in Lemma 2 is completed. □

Secondly, we prove Conclusion ② of Lemma 2 based on Conclusion ①. To render the proof process simpler, the following function is introduced

$$G(\chi_e) = \begin{cases} \dot{\chi}_d - \lambda_u(\psi)\omega_{max} - k_1 sin\chi_e, \chi_e \in [-\pi, 0) \\ \dot{\chi}_d + \lambda_u(\psi)\omega_{max} - k_1 sin\chi_e, \chi_e \in [0, \pi) \end{cases} \tag{A14}$$

The function $G(\chi_e)$ has this property: if $\chi_e \in [-\pi, 0)$, then $G(\chi_e) = -G(\chi_e + \pi)$. Thus, Conclusion ① of Lemma 2 can be summarized as $G(\chi_e) = \dot{\chi}_d + \lambda_u(\psi)\omega_{max} - k_1 sin\chi_e \geq 0$ when $\chi_e \in [0, \pi)$. Therefore, when $\chi_e \in [-\pi, 0)$ it implies $\chi_e + \pi \in [0, \pi)$, and then $G(\chi_e) = \dot{\chi}_d - \lambda_u(\psi)\omega_{max} - k_1 sin\chi_e = -G(\chi_e + \pi) \leq 0$. That means Conclusion ② of Lemma 2 is true. Thus, the proof for Lemma 2 is completed. □

## Appendix C. Proof of Lemma 3

**Proof:** The proposed heading rate control law given by Equation (24) can be rewritten as follows

$$\dot{\chi} = \begin{cases} \Omega, |\Omega| \leq \lambda_u(\psi)\omega_{max} \\ sgn(\Omega)\lambda_u(\psi)\omega_{max}, |\Omega| \geq \lambda_u(\psi)\omega_{max} \end{cases} ; \Omega = -k\lambda_u(\psi)\chi_e + \dot{\chi}_d \tag{A15}$$

Next, we prove Lemma 3 based on Lemma 2. The following trigonometric inequalities are used in the proof.

$$\begin{cases} \chi_e \geq sin\chi_e, \chi_e \in [0, \pi) \\ \chi_e \leq sin\chi_e, \chi_e \in [-\pi, 0) \end{cases} \tag{A16}$$

(a)  When $\chi_e \in [-\pi, 0)$, there are three cases.

- If $\Omega > \lambda_u(\psi)\omega_{max}$, then $\dot{\chi}_e = \lambda_u(\psi)\omega_{max} - \dot{\chi}_d \geq -k_1 sin\chi_e \geq -k_2 sin\chi_e > 0$;
- else if $\Omega < -\lambda_u(\psi)\omega_{max}$, then $\dot{\chi}_e = -\lambda_u(\psi)\omega_{max} - \dot{\chi}_d \geq -k\lambda_u(\psi)\chi_e \geq -k_2\chi_e \geq -k_2 sin\chi_e > 0$;
- else $|\Omega| \leq \lambda_u(\psi)\omega_{max}$, then $\dot{\chi}_e = -k\lambda_u(\psi)\chi_e \geq -k_2\chi_e \geq -k_2 sin\chi_e > 0$.

In summary, when $\chi_e \in [-\pi, 0)$, $\dot{\chi}_e \geq -k_2 sin\chi_e > 0$. This means that Conclusion ① of Lemma 3 is proved.

(b)  When $\chi_e \in [0, \pi)$, there are three cases.

- If $\Omega > \lambda_u(\psi)\omega_{max}$, then $\dot{\chi}_e = \lambda_u(\psi)\omega_{max} - \dot{\chi}_d \leq -k\lambda_u(\psi)\chi_e \leq -k_2 sin\chi_e < 0$;
- else if $\Omega < -\lambda_u(\psi)\omega_{max}$, then $\dot{\chi}_e = -\lambda_u(\psi)\omega_{max} - \dot{\chi}_d \leq -k_1 sin\chi_e \leq -k_2 sin\chi_e < 0$;
- else $|\Omega| \leq \lambda_u(\psi)\omega_{max}$, $\dot{\chi}_e = -k\lambda_u(\psi)\chi_e \leq -k_2 sin\chi_e < 0$.

In summary, when $\chi_e \in [0, \pi)$, $\dot{\chi}_e \leq -k_2 sin\chi_e < 0$. This means that Conclusion ② of Lemma 3 is proved. Thus, the proof for Lemma 3 is completed. □

## Appendix D. Proof of Lemma 4

**Proof:** According to Equation (9), we can obtain $\dot{r} = v_r cos(\chi - \theta) = v_r cos(\phi_{GF} + \chi_e) \leq v_r$. It can be observed from Equation (6) that $v_r \leq v_s + T$, which implies $\dot{r} \leq v_r \leq v_s + T$. Thus, it can be concluded that

$$r \leq r_0 + (v_s + T)t_c \tag{A17}$$

where $r_0$ presents the initial relative distance between the UAV and the target at time $t_0$. $t_c$ presents the time of the relative course error $\chi_e$ converging to 0.

(a)  If $|\chi_{e0}| \in [0, \frac{\pi}{2}]$, the convergence time $t_{c1}$ of process ($\chi_e \to 0$) can be obtained:

$$t_{c1} = \frac{1}{k_2} ln \left| \frac{tan\left(\frac{\chi_{e0}}{2}\right)}{tan\left(\frac{\alpha_0}{2} - \frac{\pi}{4}\right)} \right| \tag{A18}$$

Substituting (A18) into (A17), the upper bound of *r* is obtained as follows:

$$r \le r_0 + (v_s + T)\frac{1}{k_2} ln \left| \frac{tan\left(\frac{\chi_{e0}}{2}\right)}{tan\left(\frac{\alpha_0}{2} - \frac{\pi}{4}\right)} \right| \tag{A19}$$

(b)    If $|\chi_{e0}| \in \left(\frac{\pi}{2}, \pi\right]$, the relative course error $|\chi_e|$ firstly converges to $\pi/2$ in time $t_{c2}$, and then converges from $\pi/2$ to 0 in time $t_{c3}$. According to Equation (32), one has

$$t_{c2} = \frac{1}{k_2} ln \left| tan\left(\frac{\chi_{e0}}{2}\right) \right|; t_{c3} = \frac{1}{k_2} ln \left| \frac{1}{tan\left(\frac{\alpha_0}{2} - \frac{\pi}{4}\right)} \right| \tag{A20}$$

Thus, the upper bound of *r* is defined as

$$r \le r_0 + (v_s + T)\cdot(t_{c2} + t_{c3}) = r_0 + (v_s + T)\frac{1}{k_2} ln \left| \frac{tan\left(\frac{\chi_{e0}}{2}\right)}{tan\left(\frac{\alpha_0}{2} - \frac{\pi}{4}\right)} \right| \tag{A21}$$

In summary, the relative distance *r* between the drone and the target is bounded, and Lemma 4 is proved. □

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
