# Peer review of "Cooperative Standoff Target Tracking using Multiple Fixed-Wing UAVs with Input Constraints in Unknown Wind"

_drones, doi:10.3390/drones7090593_

Round 1

Reviewer 1 Report (Previous Reviewer 1)

1. The abstract of this paper described as 'In order to achieve accurate target tracking in the presence of unknown background wind and target motion', however the assumption 2 in the 4th page is written as 'The position of the moving target is assumed to be known'. 

2. The conception of airspeed that denoted as Vs in this paper is obscure. Airspeed includes True Air Speed(TAS), Indicated Air Speed(IAS), Calibrated Air Speed(CAS) or Equivalent Air Speed(EAS) in the field of aviation. Furthermore, the velocity Vs is not clearly signed In the figure (1).

Author Response

Reviewer 2 Report (Previous Reviewer 2)

The research paper appears to be a comprehensive and robust study on the challenges and solutions of cooperative standoff tracking using multiple UAVs. I'm particularly impressed by your multi-faceted approach, which not only focuses on achieving the primary objective of circular orbit tracking but also addresses crucial constraints like control input limits and external factors such as wind and target movement.

Readable, but could be improved.

Author Response

Reviewer 3 Report (Previous Reviewer 3)

Most of my previous concerns are fully addressed in the revised manuscript.

The authors should thoroughly check the English grammar/writing and any potential typos in the entire manuscript. Some of the minor typos include:

l  p.24 line 706) grammar error: "the simulation conditions in Group B are satisfy the kinematic constraints …"

l  In Figure 11 and Figure 22, there are typos in the labels of the y-axis: “SpeeAirspeed control input..”. 

Author Response

This manuscript is a resubmission of an earlier submission. The following is a list of the peer review reports and author responses from that submission.

Round 1

Reviewer 1 Report

This paper proposed two control laws that could steer multiple UAVs to fly a circular orbit around a moving target with prescribed inter-vehicle angular spacing, which based Lyapunov guidance vector field and a temporal phase. Furthermore, the paper applied an estimator to estimate the composition velocity of the unknown wind and target motion, which uses the offset vector between the UAV’s actual flight path and the desired orbit which is defined by Lyapunov guidance vector field to estimate the composition velocity. These viewpoints are innovative and practical.

The derivation process of the paper is based on the assumptions that the target follows a uniform linear motion. The transport acceleration should be considered in Fig.2 and 3 when the target moves with variable acceleration.These problems will make it difficult to ensure the convergence of the control algorithm.

Specific comments:

In the equation (8), the author should prove the Lamda >0.5.

In the equation (21), the author should give the additional remarks while r=0.

To conclude, the paper has some innovative ideas about UAVs control method in the unknown background wind and with constraint input. However, the paper should consider the time-varying speed and direct of wind and target in the simulations. 

Reviewer 2 Report

This paper presents a complex approach to standoff target tracking using multiple fixed-wing unmanned aerial vehicles (UAVs) under control input constraints. The proposed algorithms for relative range regulation and space phase separation show a unique perspective. However, there are several areas where the research could be improved.

1. Theoretical Clarity: Firstly, while the authors state that they have used a rigorous theoretical proof for their heading rate controller, this proof is not included in the manuscript, or it's not clearly highlighted. For a reader to fully appreciate the results, it is critical to understand the foundations on which these results are built. As such, it would be beneficial if the authors provided more insight into their mathematical proofs.

2. Assumption of Initial Conditions: The authors claim that the UAVs can converge to a desired circular orbit regardless of their initial position and heading. It would be useful to see a more detailed justification for this assumption. Is it possible that certain initial conditions could lead to suboptimal performance or even failures in the system? What is the impact of wind speed and direction on initial conditions?

3. Estimation Technique: The authors propose an estimator for the composition velocity of the unknown wind and target motion. However, the approach assumes that the UAV's actual flight path deviates from the desired orbit due to these effects only. This might not always be the case in real-world scenarios where other factors such as hardware imperfections or unforeseen environmental disturbances can cause deviations. The authors need to address these additional factors for a more realistic estimator.

4. Performance Evaluation: While simulation results have been presented, it would be valuable to have a comparative study with other similar algorithms. This would give more context to the results and would make the contribution of the authors clearer.

5. Implementation Feasibility: The authors should discuss the computational load and real-time feasibility of their proposed system. As we know, UAV systems are highly time-critical, and any delay in computation could render the system ineffective. This is an important aspect that the authors seem to have overlooked.

6. Robustness Evaluation: The paper lacks a thorough robustness analysis against varying parameters and conditions. It would be beneficial to see how the system behaves in extreme weather conditions, in the presence of high levels of noise, or with a sudden change in target dynamics. This would lend much more credibility to their claims of robustness.

Overall, this research provides a solution to cooperative standoff tracking, but needs to address several gaps for a more comprehensive and credible study.

The writing is okay but not elegant.

Reviewer 3 Report

* A brief summary
 This paper focuses on cooperative standoff target tracking using a team of fixed-wing UAVs and addresses challenges in input constraints (airspeed and turning rate limitation), uncertainties in background wind, and uncertainties in target motion. The proposed controller algorithm comprises (1) a heading rate control law using LGVF, (2) an airspeed control law with a concept of temporal phase, and (3) robustness to disturbances (wind and target uncertainty). The effectiveness of the proposed algorithm is evaluated through a series of simulation scenarios, comparing its performance to baseline benchmark algorithms.

* General concept comments
The manuscript is concise and well-organized. The problem definition is clear. The contributions are convincing. Simulation results seem to be reasonable. 
Publication is recommended after some improvements. 

* Specific comments
1. Based on the assumptions presented in the problem formulation, it is assumed that the position and velocity of the moving target are known or well estimated (p.4 line 193). However, the authors claim that the proposed algorithm can deal with uncertainties in target motion by regarding it as one of the external disturbances (p.4 line 170, p.5 line 216, etc.). The assumption and the contribution appear to be mismatched, which may be misleading to readers. In the description of the proposed algorithm, it is essential to clarify what kind of target information is known and what remains unknown or uncertain about the target motion. A clearer explanation is highly recommended.

2. The proposed estimation algorithm, Composition Velocity Estimation Algorithm (CVEA), only estimates the composite velocity, which combines both the target motion and wind velocity. Can the proposed algorithm estimate target motion and wind velocity separately? In practice, UAV operators may want to estimate both the current wind velocity and target velocity individually.

3. In the simulation scenario, only simple target motion (uniform linear) and stationary wind velocity (no change in wind speed and wind direction) are simulated. In my understanding, one of the main contributions of the proposed algorithm is robustness to the uncertainties on target motion and background wind. The authors are encouraged to discuss the performance of the proposed tracking/estimation algorithm under different simulation conditions (for example, target with circular/irregular/random motion, non-stationary wind direction (varying wind speed or wind direction)). 

There are some typos and grammar errors throughout the paper. It is recommended that the authors thoroughly check the whole manuscript to improve its English writing. 

 For instance, 

P.1 line 41: input constrain -> input constraints 

P.3 line 127: asymptotically convergence to -> asymptotically converge to 

P.6 line 228~233: A sentence in lines 228-229 (“The following three … control problem”) is duplicated in lines 232-233.  

P.12 line 420: is used to normalized the flight time -> is used to normalize the flight time 

P.19 line 600-601: a sentence (It implies that the … composition velocity) is weird. Please consider to re-write the sentence to be grammatically correct.